# BLEND: Behavior-guided Neural Population Dynamics Modeling via Privileged Knowledge Distillation

**Zhengrui Guo**
The Hong Kong University of Science and Technology
Beijing Institute of Collaborative Innovation
zguobc@connect.ust.hk

**Fangxu Zhou**
Peking University
zhoufangxu@pku.edu.cn

**Wei Wu**
Peking University
weiwu@stu.pku.edu.cn

**Qichen Sun**
Peking University
2000010820@stu.pku.edu.cn

**Lishuang Feng**
Beihang University
Beijing Institute of Collaborative Innovation
fenglishuang@buaa.edu.cn

**Jinzhuo Wang** ✉
Peking University
wangjinzhuo@pku.edu.cn

**Hao Chen** ✉
The Hong Kong University of Science and Technology
jhc@cse.ust.hk

## Abstract

Modeling the nonlinear dynamics of neuronal populations represents a key pursuit in computational neuroscience. Recent research has increasingly focused on jointly modeling neural activity and behavior to unravel their interconnections. Despite significant efforts, these approaches often necessitate either intricate model designs or oversimplified assumptions. Given the frequent absence of perfectly paired neural-behavioral datasets in real-world scenarios when deploying these models, a critical yet understudied research question emerges: how to develop a model that performs well using only neural activity as input at inference, while benefiting from the insights gained from behavioral signals during training?

To this end, we propose **BLEND**, the **B**ehavior-guided neura**L** population dynamics mod**E**lling framework via privileged k**N**owledge **D**istillation. By considering behavior as privileged information, we train a teacher model that takes both behavior observations (privileged features) and neural activities (regular features) as inputs. A student model is then distilled using only neural activity. Unlike existing methods, our framework is model-agnostic and avoids making strong assumptions about the relationship between behavior and neural activity. This allows BLEND to enhance existing neural dynamics modeling architectures without developing specialized models from scratch. Extensive experiments across neural population activity modeling and transcriptomic neuron identity prediction tasks demonstrate strong capabilities of BLEND, reporting over $50\%$ improvement in behavioral decoding and over $15\%$ improvement in transcriptomic neuron identity prediction after behavior-guided distillation. Furthermore, we empirically explore various behavior-guided distillation strategies within the BLEND framework and present a comprehensive analysis of effectiveness and implications for model performance. Code will be made available at https://github.com/dddavid4real/BLEND.

---

✉ : Co-corresponding Author

# 1 INTRODUCTION

Large-scale population-level recordings of neural activity enable the understanding of how complex abilities of the brain in sensing, movement, and cognition emerge from the collective activity of grouped neurons (Stevenson & Kording, 2011; Jun et al., 2017; Pei et al., 2021). This insight has led to the development of various neural dynamics modeling methods to disentangle and interpret the hidden structure of neural population activity with large-volume neural recordings as inputs (Pandarinath et al., 2018; Ye & Pandarinath, 2021; Le & Shlizerman, 2022).

Alongside the recorded neural population activity, the observed behavior signals provide crucial context and complementary information during neural dynamics modeling (Sani et al., 2021). For example, behavior allows for the integration of neural data with other physiological measures (*e.g.*, muscle activity, eye movements). By incorporating behavioral information, more comprehensive and functionally relevant models of neural dynamics have been proposed (Zhou & Wei, 2020; Hurwitz et al., 2021; Schneider et al., 2023b), bridging the gap between neural activity and its real-world manifestations.

However, neural activity recordings with paired behavior signals are not always available in real-world settings, *i.e.*, behavioral data might be partial, limited, or not available for all periods of neural recording. For instance, some studies focus on resting-state neural activity, where a subject is not engaged in any specific task or receiving external stimuli. The absence of structured tasks means there's no clear temporal alignment between neural events and behavioral events (Drew et al., 2020; Nozari et al., 2024). This discrepancy between the availability of behavioral and neural data leads to a critical distinction in the types of information accessible for model development. These features that only exist during the training stage are called *privileged features*, and those that exist throughout training and inference stages are termed *regular features* (Vapnik & Vashist, 2009). Thus, creating a model that can perform well using only regular features (neural activity) at inference time, while still benefiting from the insights gained from privileged features (behavior), represents a critical research question in neural dynamics modeling, especially in bridging the gap between controlled experimental settings and real-world applications where privileged knowledge is limited or unavailable. Maximizing the utility of existing privileged-regular feature pairs to enhance the regular-feature-based network performance is a promising strategy to answer this question. Yet in the field of computational neuroscience, these efforts remain underexplored.

To this end, in this paper, we propose **BLEND**, the **B**ehavior-guided neura**L** population dynamics mod**E**lling framework via privileged k**N**owledge **D**istillation. As shown in Fig. 1(c), BLEND constitutes a student and a teacher model, in which the teacher trains on both behavior observations and neural activity recordings, then distills knowledge to guide the student which takes only neural activity as input. This ensures the student model can make predictions using only recorded neural activity during the deployment/inference stage, but also benefits from the guidance from behavior information during the training stage, making it more versatile for settings lacking behavioral data. Our main contributions are summarized as follows:

- Built upon the privileged knowledge distillation framework, we introduce a simple yet effective neural dynamics modeling paradigm, BLEND, relying on a fundamental assumption that behavior can serve as explicit guidance for neural representation learning. Notably, our approach is model-agnostic, allowing for its seamless integration with diverse existing neural dynamics modeling architectures, thus avoiding the need for developing specialized models from scratch.

- To evaluate our framework, we conduct extensive experiments on two benchmarks, *i.e.*, Neural Latents Benchmark'21 for neural activity prediction, behavior decoding, and matching to peri-stimulus time histograms (PSTHs), as well as a multi-modal calcium imaging dataset for transcriptomic identity prediction. Results show that our framework elevates the performance of baseline methods by a large margin ($>50\%$ improvement in behavioral decoding and $>15\%$ improvement in neuronal identity prediction) and significantly outperforms the state-of-the-art (SOTA) models.

- We present a comprehensive analysis of our framework across base models and privileged knowledge distillation strategies, revealing key insights into the interaction between neural activity and behavior. Our results demonstrate that behavior-guided distillation not only improves model performance but fundamentally enhances the quality of learned neural representations. This leads to more accurate and nuanced modeling of neural dynamics, offering new perspectives on how behavioral information can be leveraged to better understand complex neural patterns.

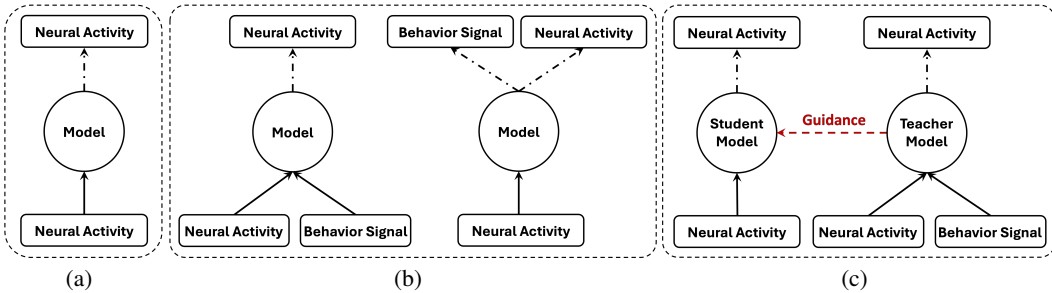

Figure 1: Schematic illustration of neural population dynamics modeling mechanisms. In this paper, we benchmark all the methods under the framework of masked neural activity reconstruction, in which the model is firstly trained in an unsupervised manner to reconstruct the randomly masked neural activity and then applied to downstream tasks such as neural activity prediction and behavior decoding. (a) Neural dynamics modeling methods that only use neural population activity as input. (b) Neural dynamics modeling methods that take behavior information as a prior. (c) Our BLEND framework, which considers behavior information as privileged knowledge for distillation.

## 2 RELATED WORK

**Neural dynamics modeling (NDM).** NDM refers to a category of methods that aim to capture the dynamics of neural activity by using the activity recordings as inputs. A common and effective choice is the latent variable model (LVM), which leverages low-dimensional latent factors to interpret these dynamics. Various LVMs are developed, ranging from simple non-temporal models such as principal components analysis (PCA) (Cunningham & Yu, 2014) and its variants (Kobak et al., 2016), to linear dynamical systems (Macke et al., 2011; Gao et al., 2016), and to complex state space models like LFADS (Pandarinath et al., 2018). Since the advent of Transformer, its ability to capture temporal dependencies and long-range interactions makes it appropriate for neural data, with representative works such as NeuralDataTransformer (NDT) (Ye & Pandarinath, 2021), STNDT (Le & Shlizerman, 2022), and EIT (Liu et al., 2022). As shown in Fig. 1(a), methods in this category purely depend on neural activity recordings for neural population dynamics modeling yet ignore using the paired behavior information as guidance.

**Neural dynamics modeling with behavior as a prior.** In recent years, an emerging research direction has focused on jointly modeling neural population dynamics and behavioral signals. As illustrated in Fig. 1(b), these methods can be categorized into two types. The first inputs both behavior and neural activity and utilizes behavior signals to guide the learning of neural dynamics: pi-VAE (Zhou & Wei, 2020) considers the behavior variables as constraints for the construction of latent space of LVMs; CEBRA (Schneider et al., 2023b) utilizes behavior signals to construct contrastive learning samples for label-informed neural activity analysis. However, these approaches normally need to develop delicately designed modules or complex training strategies to achieve their goal. The second line of work aims to decompose neural activity into behavior-relevant and behavior-irrelevant dynamics and reconstruct both behavior signals and neural activity signals. To that end, a linear state-space model, PSID (Sani et al., 2021), is developed to decode population dynamics from motor brain regions. Nonlinear state-space models such as TNDM (Hurwitz et al., 2021) and SABLE (Jude et al., 2022) are proposed to further capture the nonlinear dynamics of neural activity and behavior observation. Nonetheless, these models assume a clear distinction between behaviorally relevant and irrelevant dynamics in input neural activity, which might not always be practical, potentially leading to oversimplification.

In contrast, our approach offers a model-agnostic learning paradigm that can be directly applied to existing LVM models without relying on strong assumptions, thereby circumventing the issues present in current LVM models that integrate behavior information as a prior.

**Learning under privileged information (LUPI).** Firstly proposed in Vapnik & Vashist (2009), LUPI refers to the setting where, alongside the primary data modality, the model has access to an additional source of information. This extra source of input, termed privileged, is exclusively available during the training phase. The main objective of LUPI is to leverage this privileged information to learn a better model in the primary data modality than one would learn without the privileged information. This learning paradigm has been applied to different machine learning problems such as recommendation system (Yang et al., 2022), medical image analysis (Chen et al., 2021), emotion recognition (Aslam et al., 2023), and semantic segmentation (Liu et al., 2024).

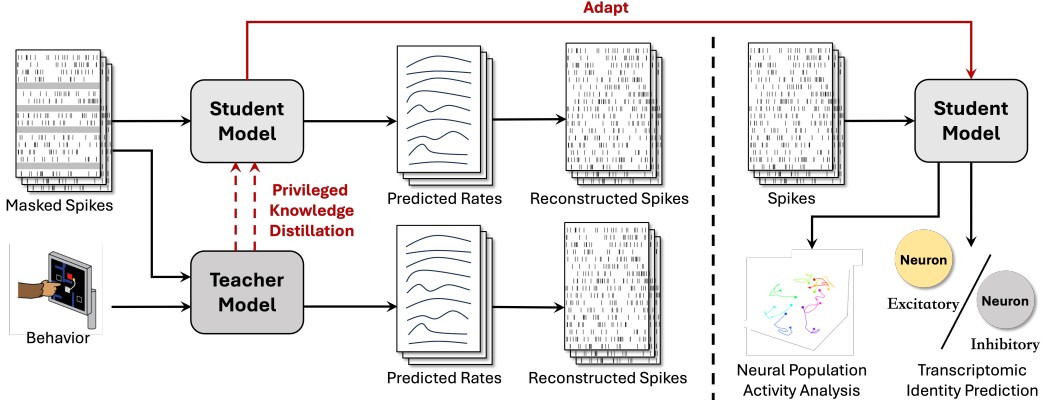

Figure 2: Illustration of the proposed **BLEND** framework, exemplified using neural spiking activity data. **Left:** Behavior-guided neural representation learning via privileged knowledge distillation. The teacher model is trained on a composite of neural activity and behavioral signals, subsequently distilling its knowledge to a student model that utilizes solely neural activity as input. **Right:** During the inference phase, the distilled student model is employed for neural population activity analysis and transcriptomic identity prediction tasks.

In computational neuroscience, considering behavior information as privileged information to guide neural dynamics modeling remains understudied. This work develops a novel learning paradigm that represents an inaugural step toward addressing this underexplored research question and advancing the field.

## 3 METHODS

This section introduces the details of our proposed **BLEND** framework, *i.e.*, the behavior-guided neural population dynamics modeling via privileged knowledge distillation. We start with problem formulation of behavior-guided neural dynamics modeling in Section 3.1, followed by an exposition of the BLEND algorithm in Section 3.2. Detailed exploration of the effectiveness of the BLEND framework can be found in Appendix A.8 and A.9, including empirical studies of implications of behavior guidance and how to choose distillation strategies for different models and data.

### 3.1 PROBLEM FORMULATION

We concretize the learning problem in this study with neural spiking data. For each trial of the input neural activity, let $\mathbf{x} \in \mathcal{X} = \mathbb{N}^{N \times T}$ represent the input spike counts, where $\mathbf{x}_i^t$ denotes the spike count for neuron $i$ at time $t$. Let $\mathbf{b} \in \mathcal{B} = \mathbb{R}^{B \times T}$ be the corresponding behavior signal, with $\mathbf{b}_i^t$ denoting the behavioral signal at time $t$. Without loss of generality, we assume that behavioral signals are temporally continuous and have the same number of time steps as the neural activity.[1] As illustrated in Fig. 1, the model $f_\theta$ aims to reconstruct the randomly masked neural activity solely based on the unmasked portions of $\mathbf{x}$ or together with the auxiliary behavior signal $\mathbf{b}$. The optimization objective for masked time-series modeling (MTM) can be formulated as follows:

$$\theta^* = \arg\min_{\theta} \mathbb{E}_{\mathbf{x} \sim p(\mathbf{x}), \mathbf{b} \sim p(\mathbf{b}), \mathbf{m} \sim p(\mathbf{m})} \left[ \frac{1}{|\mathbf{m}|} \sum_{i,t} \mathbf{m}_i^t \cdot \mathcal{L}_{\text{rec}} \left( f_\theta \left( \mathbf{x}_{\bar{m}}, \mathbf{b} \right)_i^t, \mathbf{x}_i^t \right) \right] \triangleq \mathcal{L}_{\text{MTM}} \quad (1)$$

where $\mathbf{m} \in \{\mathbf{0}, \mathbf{1}\}^{N \times T}$ is a binary mask with $\mathbf{1}$ indicating masked elements, $\mathbf{x}_{\bar{m}} \in \mathcal{X}_{\bar{m}}$ represents the unmasked portions of $\mathbf{x}$, $|\mathbf{m}|$ is the number of masked elements, and $\mathcal{L}_{\text{rec}}$ is either Poisson or Cross-Entropy loss as specified in the model configuration.

### 3.2 BLEND

The goal of this work is to create a model that can make predictions using only neural activity data during deployment/inference, but also benefits from behavior data during training. This subsection details the algorithm of BLEND, including the formulation of privileged knowledge distillation under the context of neural dynamics modeling, teacher-student architecture, and distillation strategies.

---

[1]BLEND can be readily extended to accommodate scenarios wherein behavioral information is discrete or non-temporal in nature, see preliminary explorations in Appendix A.10.

We first conceptualize the behavior-guided neural population dynamics modeling problem under the framework of LUPI:

**Definition 1** (Privileged Knowledge (Yang et al., 2022)). *Consider a general learning problem with input* $\mathbf{x} \in \mathcal{X}$ *and label* $\mathbf{y} \in \mathcal{Y}$. *If there is an additional source of information* $\mathbf{b} \in \mathcal{B}$ *that exists during training but not inference, we say* $\mathbf{b}$ *is the **privileged knowledge** if and only if* $I(\mathbf{y}; \mathbf{b}|\mathbf{x}) := H(\mathbf{y}|\mathbf{x}) - H(\mathbf{y}|\mathbf{x}, \mathbf{b}) > 0$.

In Definition 1, $I(\cdot|\cdot)$ and $H(\cdot|\cdot)$ represent conditional mutual information and conditional entropy, respectively. According to Definition 1, the behavior information $\mathbf{b}$ serves as the privileged knowledge and offers additional predictive capabilities of label $\mathbf{y}$, which is the unmasked neural activity $\mathbf{x}$ in our scenario. Next, we introduce the proposed privileged knowledge distillation framework, which is designed to incorporate behavioral guidance and facilitate neural dynamics modeling.

### 3.2.1 PRIVILEGED KNOWLEDGE DISTILLATION WITH BEHAVIOR AS GUIDANCE

As is illustrated in Fig. 2, BLEND employs a teacher-student architecture to realize our goal: by leveraging the privileged information $\mathbf{b}$ at the training phase, to learn an LVM model for inference phase that outperforms those built on the regular feature $\mathbf{x}$ alone. This procedure is divided into two sequential stages:

**Stage 1: Train a teacher model with regular information and privileged information.** The teacher model $f_{\theta_T} \in \{f | f : \mathcal{X}_{\bar{m}} \times \mathcal{B} \mapsto \mathcal{X}\}$ takes both masked neural activity recordings $\mathbf{x}$ and the corresponding behavior signals $\mathbf{b}$ as inputs. The model parameter $\theta_T$ is optimized by minimizing the MTM loss $\mathcal{L}_{\text{MTM}}$ formulated in Eq. (1).

**Stage 2: Train a student model with regular information by distillation.** By exploiting all available sources of information, the teacher model acquires a rich understanding of the neural population dynamics. We then aim to distill the learned knowledge to the student model $f_{\theta_S} \in \{f | f : \mathcal{X}_{\bar{m}} \mapsto \mathcal{X}\}$, which takes the neural activity as input. The parameter of student model $\theta_S$ is updated by minimizing the following objective:

$$
\mathbb{E}_{\mathbf{x}, \mathbf{b}, \mathbf{m}} \left[ \alpha \cdot \underbrace{\frac{1}{|\mathbf{m}|} \sum_{i,t} \mathbf{m}_i^t \cdot \mathcal{L}_{\text{rec}} \left( f_{\theta_S} \left( \mathbf{x}_{\bar{m}} \right)_i^t, \mathbf{x}_i^t \right)}_{\text{MTM Loss}} + (1 - \alpha) \cdot \underbrace{\sum_{i,t} \mathcal{L}_{\text{distill}} \left( f_{\theta_S} \left( \mathbf{x}_{\bar{m}} \right)_i^t, f_{\theta_T} \left( \mathbf{x}_{\bar{m}}, \mathbf{b} \right)_i^t \right)}_{\text{Distillation Loss}} \right],
$$

$$(2)$$

where $\alpha \in (0, 1)$ denotes the mixing ratio between the MTM loss and the distillation loss. The entire optimization procedure of the proposed BLEND framework is presented in Alg. 1. After training the student model with privileged knowledge distillation for MTM, it is applied to downstream tasks where only neural activity recordings are available for inference (shown in Fig. 2, right).

While the above algorithm outlines the general framework of BLEND, different implementations of the distillation loss $\mathcal{L}_{\text{distill}}$ can capture various aspects of the teacher's knowledge. In this study, we investigate the following four main distillation strategies in our BLEND framework, each designed to transfer certain aspects of the teacher's knowledge to the student model.

**Hard distillation.** As a baseline, we first implement the $\mathcal{L}_{\text{distill}}$ with hard distillation strategy, which directly minimizes the mean squared error (MSE) between the teacher and student outputs:

$$
\mathcal{L}_{\text{distill}} \left( f_{\theta_S} \left( \mathbf{x}_{\bar{m}} \right)_i^t, f_{\theta_T} \left( \mathbf{x}_{\bar{m}}, \mathbf{b} \right)_i^t \right) = \left\| f_{\theta_S} \left( \mathbf{x}_{\bar{m}} \right)_i^t - f_{\theta_T} \left( \mathbf{x}_{\bar{m}}, \mathbf{b} \right)_i^t \right\|_2^2 \tag{3}
$$

**Soft distillation.** This approach distills knowledge by matching the softened probability distributions of the teacher and student models. We use a temperature parameter $\tau$ to soften the logits before applying the softmax function:

$$
\mathcal{L}_{\text{distill}} \left( f_{\theta_S} \left( \mathbf{x}_{\bar{m}} \right)_i^t, f_{\theta_T} \left( \mathbf{x}_{\bar{m}}, \mathbf{b} \right)_i^t \right) = \tau^2 \cdot \text{KL} \left( \sigma \left( \frac{f_{\theta_T} (\mathbf{x}_{\bar{m}}, \mathbf{b})_i^t}{\tau} \right) \middle\| \sigma \left( \frac{f_{\theta_S} (\mathbf{x}_{\bar{m}})_i^t}{\tau} \right) \right), \tag{4}
$$

where $\sigma$ is the softmax function, $\text{KL}(\cdot \| \cdot)$ is the Kullback-Leibler divergence, and $\tau$ is the temperature parameter.

**Feature distillation.** This method aims to align all the intermediate representations of the student model with those of the teacher model.

$$\mathcal{L}_{\text{distill}} \left( f_{\theta_S} \left( \mathbf{x}_{\bar{m}} \right)_i^t, f_{\theta_T} \left( \mathbf{x}_{\bar{m}}, \mathbf{b} \right)_i^t \right) = \sum_{l=1}^{L} \left\| f_{\theta_S}^l (\mathbf{x}_{\bar{m}})_i^t - f_{\theta_T}^l (\mathbf{x}_{\bar{m}}, \mathbf{b})_i^t \right\|_2^2, \qquad (5)$$

where $f_{\theta_S}^l$ and $f_{\theta_T}^l$ denote the outputs of the $l$-th layer of the student and teacher models, respectively.

**Correlation distillation.** This approach focuses on preserving the correlation structure of the teacher's outputs in the student model. For each sample in the batch, we compute the correlation matrices of the teacher and student outputs and minimize their difference:

$$\mathcal{L}_{\text{distill}} \left( f_{\theta_S} \left( \mathbf{x}_{\bar{m}} \right)_i^t, f_{\theta_T} \left( \mathbf{x}_{\bar{m}}, \mathbf{b} \right)_i^t \right) = \frac{1}{B} \sum_{j=1}^{B} \left\| \text{Corr}(f_{\theta_S}(\mathbf{x}_{\bar{m}})_j) - \text{Corr}(f_{\theta_T}(\mathbf{x}_{\bar{m}}, \mathbf{b})_j) \right\|_2^2, \qquad (6)$$

where $B$ denotes the batch size, and $\text{Corr}(\cdot)$ computes the correlation matrix of the outputs for the $j$-th sample in the batch (see Appendix A.1.2 for details).

## 4 EXPERIMENTS

As shown in Fig. 2, this work includes two benchmarks for evaluating our proposed BLEND framework. The first is a public benchmark for neural latent dynamics model evaluation from Pei et al. (2021), named Neural Latents Benchmark'21 (NLB'21). The second is a recent, public multi-modal neural dataset from Bugeon et al. (2022), which contains calcium imaging recordings of neural population activity as well as single-cell spatial transcriptomics of the recorded tissue. Details of these two benchmarks, tasks, and included baselines are introduced in Section 4.1 and 4.2.

### 4.1 NEURAL POPULATION ACTIVITY ANALYSIS ON NLB'21 BENCHMARK

We include three sub-datasets from NLB'21, *i.e.*, MC-Maze, MC-RTT, and Area2-Bump, for a series of neural dynamics modeling evaluations. These datasets contain neural recordings from monkeys performing various reaching tasks: MC-Maze features delayed reaches through virtual mazes, MC-RTT involves continuous reaches to random targets without delays, and Area2-Bump includes reaches with occasional mechanical perturbations (see Appendix A.2 for details). To evaluate the capability of our proposed BLEND on neural population dynamics modeling, we adopt three tasks from NLB'21, *i.e.*, neural activity prediction, behavior decoding, and matching to peri-stimulus time histograms (PSTHs).

**Neural activity prediction** task aims to predict the neural activity of held-out neurons, measured by metric *Co-bps*, computed as the log-likelihood of held-out neurons' activity (see Appendix A.3 for detailed computation steps).

**Behavior decoding** task requires the model to relate neural activity to observed behavior. To evaluate this task, we first train a ridge regression model to predict behavioral data from neural firing rates in the training set. We then use this model to predict behavior from neural activity in the test set and measure the accuracy of these predictions using the $R^2$ score (named Vel-$R^2$).

**Match to PSTHs** task aims to evaluate how well models can capture stereotyped features of neuronal responses across repeated trials of the same condition. To evaluate it, we calculate the $R^2$ between model-predicted trial-averaged rates and true PSTHs for each neuron across all conditions, then average these $R^2$ values across neurons (named PSTH-$R^2$).

We include two types of models as baselines for this benchmark. The first solely uses neural activity recordings as inputs, including LRNN (Stolzenburg et al., 2018), LFADS (Pandarinath et al., 2018), NeuralDataTransformer (NDT) (Ye & Pandarinath, 2021), STNDT (Le & Shlizerman, 2022), and MINT (Perkins et al., 2023). The second type utilizes behavior information during training as a prior to facilitate the learning process of neural dynamics modeling, including pi-VAE (Zhou & Wei, 2020), PISD (Sani et al., 2021), and TNDM (Hurwitz et al., 2021). We choose NDT and LFADS as the base model for privileged knowledge distillation using behavior information, with best-distilled model performance reported in Tab. 1. See details of model configurations in Appendix A.4.

Table 1: Comparison of the proposed **BLEND framework** with other neural dynamics modeling methods on NLB'21 Benchmark (Pei et al., 2021). **Bold values** denote the best performance for the corresponding metric.

| Methods | MC-Maze | | | MC-RTT | Area2-Bump | |
|---|---|---|---|---|---|---|
| | Co-bps | Vel-$R^2$ | PSTH-$R^2$ | Vel-$R^2$ | Vel-$R^2$ | PSTH-$R^2$ |
| **Neural Decoding** | | | | | | |
| LRNN (Stolzenburg et al., 2018) | 0.148 | 0.317 | 0.274 | 0.188 | 0.473 | 0.085 |
| MINT (Perkins et al., 2023) | 0.181 | 0.646 | 0.165 | 0.159 | 0.370 | 0.103 |
| STNDT (Le & Shlizerman, 2022) | 0.282 | 0.773 | 0.585 | 0.233 | 0.563 | 0.416 |
| **Behavior as Prior** | | | | | | |
| pi-VAE (Zhou & Wei, 2020) | 0.214 | 0.621 | 0.455 | 0.265 | 0.434 | 0.305 |
| RNN PSID (Sani et al., 2021) | 0.229 | 0.683 | 0.499 | 0.287 | 0.514 | 0.372 |
| TNDM (Hurwitz et al., 2021) | 0.248 | 0.730 | 0.509 | 0.345 | 0.677 | 0.402 |
| **BLEND** | | | | | | |
| NDT (Ye & Pandarinath, 2021) | 0.275 (+0.0%) | 0.779 (+0.0%) | 0.551 (+0.0%) | 0.318 (+0.0%) | 0.519 (+0.0%) | 0.290 (+0.0%) |
| NDT-Distill-Best (Ours) | 0.310 (+12.7%) | **0.891 (+14.4%)** | 0.592 (+7.4%) | 0.372 (+17.0%) | 0.788 (+51.8%) | 0.483 (+66.6%) |
| LFADS (Pandarinath et al., 2018) | 0.315 (+0.0%) | 0.858 (+0.0%) | 0.579 (+0.0%) | 0.416 (+0.0%) | 0.649 (+0.0%) | 0.425 (+0.0%) |
| LFADS-Distill-Best (Ours) | **0.321 (+1.9%)** | 0.877 (+2.2%) | **0.604 (+4.3%)** | **0.429 (+3.1%)** | **0.837 (+29.0%)** | **0.615 (+44.7%)** |

## 4.2 TRANSCRIPTOMIC NEURON IDENTITY PREDICTION ON MULTI-MODAL NEURAL ACTIVITY DATASET

This multi-modal dataset combines calcium imaging from mouse primary visual cortex (V1) with single-cell transcriptomics. The dataset includes functional recordings from 9728 neurons across 17 sessions from 4 mice (SB025, SB026, SB028, SB030). Additionally, the dataset provides transcriptomic profiles that are used to categorize neurons as either excitatory or inhibitory. For mouse SB025, about half of the neurons classified as inhibitory are further subdivided into specific subtypes (see details of this dataset in Appendix A.2). We evaluate models on the following two tasks under both multi-animal scenario (all 4 mice) and single-animal scenario (only mouse SB025):

**Excitatory/inhibitory (EI) neuron identity prediction** task is a binary classification task that requires the model to predict the neuron identity, *i.e.*, excitatory or inhibitory.

**Subclass label prediction** task is a 5-class classification task, where the model is required to predict the subclass label of inhibitory neurons, *i.e.*, Lamp5, Pvalb, Vip, Sncg, or Sst.

Following the settings in Mi et al. (2023), we include a random model, PCA (Cunningham & Yu, 2014), UMAP (McInnes et al., 2018), LOLCAT (Schneider et al., 2023a) and its variants, and NeuPRINT (Mi et al., 2023) as the baseline models for this benchmark. We choose NeuPRINT as the base model for privileged knowledge distillation using behavior information with best-distilled model performance reported in Tab. 2. Detailed model configurations are in Appendix A.4.

## 5 RESULTS AND ANALYSIS

### 5.1 BENCHMARK 1: NEURAL POPULATION ACTIVITY ANALYSIS

Tab. 1 summarizes the results of neural activity prediction, behavior decoding, and match to PSTHs tasks of baseline methods and our proposed method on MC-Maze, MC-RTT, and Area2-Bump datasets. Our behavior-guided distilled NDT and LFADS models achieve the best performance and outperform SOTA methods on all tasks and datasets, demonstrating the effectiveness of our proposed BLEND mechanism.

**Fit to neural activity**. For neural activity prediction of held-out neurons, NDT-Distill-Best achieves 0.310 Co-bps score on the MC-Maze dataset, reporting 12.7% improvement over the base model NDT. LFADS-Distill-Best achieves a 0.321 Co-bps score, reporting the best performance for this task against other baseline models. These observations indicate that our models gain a better understanding of neural population dynamics and yield improved predictions of held-out neural activity.

**Fit to behavior**. For behavior (monkey hand velocity) decoding, NDT-Distill-Best achieves Vel-$R^2$ of 0.891, 0.372, and 0.788 respectively, outperforming the compared baselines. A 51.8% improvement is observed on the Area2-Bump dataset over the base model NDT. While for LFADS-Distill-Best, Vel-$R^2$ scores of 0.877, 0.429, and 0.837 are reported, showcasing better behavior decoding capabilities over baselines and the distilled NDT. Our findings demonstrate that the proposed method achieves a superior correlation between neural activity and behavioral signals compared to existing models. Note that the improvement of distilled LFADS over the base model on MC-Maze and MC-

Table 2: Top-1 accuracy of excitatory/inhibitory neuron identity prediction and inhibitory neuron subclass label prediction tasks on the multimodal population activity dataset from Bugeon et al. (2022). **Bold values** denote the best performance for the corresponding metric.

| Methods | Multiple Animals | | Single Animal | |
|---|---|---|---|---|
| | EI (2 class) | Subclass (5 class) | EI (2 class) | Subclass (5 class) |
| **Random** | 0.523 | 0.302 | 0.488 | 0.260 |
| **PCA (Cunningham & Yu, 2014)** | 0.565 | 0.330 | 0.572 | 0.263 |
| **UMAP (McInnes et al., 2018)** | 0.520 | 0.340 | 0.438 | 0.281 |
| **LOLCAT (Schneider et al., 2023a)** | 0.600 | 0.404 | 0.608 | 0.474 |
| **LOLCAT$_{ISI}$(Mi et al., 2023)** | 0.640 | 0.474 | 0.632 | 0.491 |
| **LOLCAT$_{Raw}$(Mi et al., 2023)** | 0.664 | 0.439 | 0.616 | 0.386 |
| **NeuPRINT (Mi et al., 2023)** | 0.748 (+0.0%) | 0.495 (+0.0%) | 0.667 (+0.0%) | 0.508 (+0.0%) |
| **NeuPRINT-Distill-Best (Ours)** | **0.789 (+5.5%)** | **0.571 (+15.4%)** | **0.722 (+8.2%)** | **0.588 (+15.7%)** |

RTT is not as significant as distilled NDT. The potential reason behind this is that LFADS, being RNN-based, may reach its capacity limits with larger datasets, making it harder to incorporate additional knowledge effectively. Meanwhile, LFADS performs dimensionality reduction as part of its model, which might limit its ability to incorporate additional information from the teacher in larger datasets where the intrinsic dimensionality is already well-captured.

**Match to PSTHs**. In the PSTH matching task, our distilled NDT attains PSTH-$R^2$ scores of $0.592$ and $0.483$ on the MC-Maze and Area2-Bump datasets, respectively. These results represent improvements of 7.4% and a remarkable 66.6% over the baseline model performance. The distilled LFADS model achieves PSTH-$R^2$ scores of $0.604$ and $0.615$, demonstrating superior performance among all evaluated models and yielding improvements of 4.3% and 44.7% over the base model, respectively. These findings indicate that leveraging behavioral data as privileged knowledge enhances the models' capacity to reproduce stereotyped neural responses, as quantified by PSTHs.

Overall, the BLEND framework provides a simple yet effective method that could directly improve the performance of existing NDM models in neural population activity analysis tasks.

## 5.2 BENCHMARK 2: TRANSCRIPTOMIC NEURON IDENTITY PREDICTION

Tab. 2 shows the experimental results of transcriptomic identity prediction tasks, including EI neuron identity prediction and inhibitory neuron subclass label prediction. These two tasks are conducted in both multi-animal and single-animal settings. By incorporating our BLEND framework, the distilled model shows remarkable improvement over the base model and outperforms all baseline methods.

**EI prediction**. In the excitatory/inhibitory neuron identity prediction task, our distilled NeuPRINT model attains the top-1 accuracy of $0.789$ and $0.722$ in multi-animal and single-animal settings, respectively. These findings demonstrate improvements of $5.5\%$ and $8.2\%$ over the base model. Notably, our proposed method exhibits superior performance, surpassing all other baseline approaches included for comparison.

**Subclass label prediction**. For the inhibitory neuron subclass label prediction task, the distilled NeuPRINT model improves the base model by $15.4\%$ and $15.7\%$ in multi-animal and single-animal settings, respectively, achieving the best top-1 accuracy of $0.571$ and $0.588$. Given that this task involves a five-class classification problem, the model's performance metrics are comparatively lower than those observed in the EI prediction task, which is a binary classification problem.

As illustrated in Table 2, the model's performance metrics in the single-animal setting are comparatively lower than those observed in the multi-animal setting. We posit that the multi-animal scenario provides a more diverse and extensive set of neural activity inputs, thereby enhancing the model's generalization capabilities. Besides, Mi et al. (2023) points out a data-limited regime in this benchmark characterized by scarce labeled data for transcriptomic prediction, with only a small fraction of neurons having both calcium imaging and transcriptomic data, and an imbalanced distribution across subclasses. Our BLEND framework alleviates this problem and improves the model performance since the teacher model provides regularization and rich knowledge transfer to the student.

Table 3: Ablation study on privileged knowledge distillation strategies evaluated by neural dynamics modeling tasks of NLB'21 Benchmark (Pei et al., 2021) and cell class prediction tasks of Bugeon et al. (2022) dataset. **Bold values** denote the best performance for the corresponding metric.

| **Neural Population Activity Analysis** | | | | | | |
|---|---|---|---|---|---|---|
| **Methods** | **MC-Maze** | | | **MC-RTT** | **Area2-Bump** | |
| | Co-bps | Vel-$R^2$ | PSTH-$R^2$ | Vel-$R^2$ | Vel-$R^2$ | PSTH-$R^2$ |
| **NDT (Ye & Pandarinath, 2021)** | 0.275 (+0.0%) | 0.779 (+0.0%) | 0.551 (+0.0%) | 0.318 (+0.0%) | 0.519 (+0.0%) | 0.290 (+0.0%) |
| **NDT-Hard-Distill (Ours)** | 0.290 (+5.5%) | 0.890 (+14.2%) | 0.587 (+6.5%) | 0.380 (+19.5%) | 0.747 (+43.9%) | 0.363 (+25.2%) |
| **NDT-Soft-Distill (Ours)** | 0.259 (−5.8%) | 0.881 (+13.1%) | 0.491 (−10.9%) | 0.355 (+11.6%) | 0.744 (+43.4%) | 0.318 (+9.7%) |
| **NDT-Feature-Distill (Ours)** | 0.303 (+10.2%) | 0.888 (+14.0%) | 0.596 (+8.2%) | 0.381 (+19.8%) | 0.787 (+51.6%) | 0.481 (+65.9%) |
| **NDT-Correlation-Distill (Ours)** | 0.310 (+12.7%) | **0.891 (+14.4%)** | 0.592 (+7.4%) | 0.372 (+17.0%) | 0.788 (+51.8%) | 0.483 (+66.6%) |
| **LFADS (Pandarinath et al., 2018)** | 0.315 (+0.0%) | 0.858 (+0.0%) | 0.579 (+0.0%) | 0.416 (+0.0%) | 0.649 (+0.0%) | 0.425 (+0.0%) |
| **LFADS-Hard-Distill (Ours)** | 0.321 (+1.9%) | 0.877 (+2.2%) | **0.604 (+4.3%)** | 0.429 (+3.1%) | **0.837 (+29.0%)** | **0.615 (+44.7%)** |
| **LFADS-Soft-Distill (Ours)** | 0.282 (−10.5%) | 0.800 (−6.8%) | 0.516 (−10.9%) | **0.443 (+6.5%)** | 0.740 (+14.0%) | 0.576 (+35.5%) |

| **EI Neuron Identity & Inhibitory Neuron Subclass Label Prediction** | | | | |
|---|---|---|---|---|
| **Methods** | **Multiple Animals** | | **Single Animal** | |
| | EI (2 class) | Subclass (5 class) | EI (2 class) | Subclass (5 class) |
| **NeuPRINT (Mi et al., 2023)** | 0.748 (+0.0%) | 0.495 (+0.0%) | 0.667 (+0.0%) | 0.508 (+0.0%) |
| **NeuPRINT-Hard-Distill (Ours)** | 0.789 (+5.5%) | **0.571 (+15.4%)** | 0.714 (+7.0%) | 0.581 (+14.4%) |
| **NeuPRINT-Soft-Distill (Ours)** | 0.782 (+4.5%) | 0.541 (+9.3%) | **0.722 (+8.2%)** | **0.588 (+15.7%)** |
| **NeuPRINT-Feature-Distill (Ours)** | 0.781 (+4.4%) | 0.531 (+7.3%) | 0.718 (+7.6%) | 0.524 (+3.6%) |
| **NeuPRINT-Correlation-Distill (Ours)** | **0.792 (+5.9%)** | 0.557 (+12.5%) | 0.705 (+5.7%) | 0.570 (+12.2%) |

### 5.3 ABLATION STUDIES ON BEHAVIOR-GUIDED KNOWLEDGE DISTILLATION STRATEGIES

In this work, we explore different privileged knowledge distillation strategies, including hard distillation, soft distillation, feature distillation, and correlation distillation, as detailed in Section 3.2.1.

**Neural population activity analysis on NLB'21 benchmark**. We find that the NDT-Correlation-Distill model generally performs best across most metrics, showing significant improvements over the baseline NDT. For MC-Maze, NDT-Correlation-Distill achieves the highest Vel-$R^2$ (0.891). While for LFADS and its distilled variants, LFADS-Hard-Distill shows improvements over the baseline LFADS, particularly in PSTH-$R^2$ for MC-Maze and Vel-$R^2$ for Area2-Bump. Notably, except for LFADS-Hard-Distill, other distillation methods lead to performance degradation for LFADS on MC-Maze (hence, only results for Soft-Distill are presented in Table 3). This contrasts with the NDT results, where multiple distillation strategies show improvements, indicating that not all knowledge transfer techniques are universally beneficial and highlighting the importance of careful method selection when applying distillation techniques to different models or tasks.

**Transcriptomic identity prediction on multi-modal calcium imaging dataset**. It can be seen that NeuPRINT-Hard-Distill and NeuPRINT-Soft-Distill show consistent improvements over the baseline NeuPRINT across all tasks. NeuPRINT-Correlation-Distill performs best for Multiple Animals EI prediction. NeuPRINT-Soft-Distill excels in Single Animal predictions for both EI and Subclass.

Overall, the distillation strategies, particularly Hard-Distill and Correlation-Distill, tend to outperform their respective baselines, suggesting that privileged knowledge distillation can significantly enhance neural dynamics modeling. Different distillation strategies seem to be more effective for different tasks and metrics, indicating that the choice of distillation method should be task-specific.

### 5.4 QUALITATIVE ANALYSIS

We visualized the decoded behavior trajectories of the base model and our distilled model for comparison in Fig. 3. Subfigure (a) shows the comparison between predicted and ground truth 2D hand movement on MC-Maze for NDT (base model) and NDT-Hard-Distill (our distilled model). And subfigure (b) illustrates the 1D hand velocity over time for both the X and Y axes separately. For a more comprehensive visualization and qualitative analysis, please refer to Appendix A.5 and A.6.

**2D hand movement trajectory decoding**. For NDT the base model, the predicted trajectory follows the general U-shape of the ground truth but shows significant deviations, especially at the bottom of the U and the upper right corner. While our distilled model generates a trajectory that aligns much

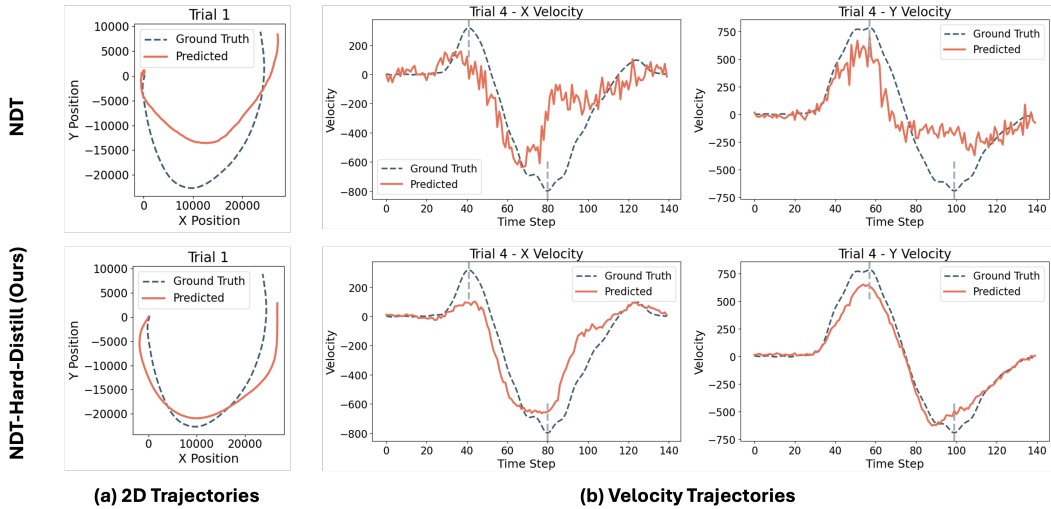

Figure 3: Visualization of behavior decoding on MC-Maze dataset. (a) Prediction and ground truth of 2D hand movement trajectory. (b) Prediction and ground truth of X and Y velocities, respectively.

more closely with the ground truth. It captures the U-shape more accurately, especially at the bottom curve and the endpoints, showing a marked improvement in behavior decoding.

**1D hand velocity decoding (X and Y axes)**. For both X and Y velocities, the base model struggles to accurately capture the magnitude and timing of velocity changes in both dimensions. Specifically, it underestimates peak velocities and shows erratic fluctuations in X velocity, and shows substantial deviations from ground truth in Y velocity, particularly after time step 60. While our distilled NDT significantly enhances velocity predictions in both X and Y directions. It more accurately captures the magnitude, timing, and overall pattern of velocity changes with improved smoothness, suggesting a better understanding of the hand movement dynamics.

## 6 CONCLUSION AND DISCUSSION

In this study, we introduce BLEND, a novel framework for neural population dynamics modeling that leverages privileged knowledge distillation. Predicated on the fundamental premise that behavioral data can serve as an explicit guide for neural representation learning, we present a model-agnostic framework that facilitates seamless integration with diverse existing neural dynamics modeling architectures. This versatility enables the enhancement of a wide range of computational approaches in the field of computational neuroscience. Comprehensive empirical evaluations, encompassing neural population activity analysis and transcriptomic identity prediction tasks, substantiate the superior effectiveness of our proposed framework. BLEND demonstrates a remarkable capacity for capturing implicit patterns within neural activity, consistently outperforming state-of-the-art models by a significant margin, thereby providing new perspectives on how behavioral observations can be leveraged to guide the complex modeling of neuronal population dynamics.

Although the initial results of BLEND are promising, a few limitations remain to be addressed. In this work, we employ a straightforward approach to implement the behavior guidance by concatenating the neural activity and behavior information along the feature dimension (assuming they have the same length of time steps). This methodological choice is motivated by the relatively low-dimensional nature of behavioral signals, which precludes their use as an independent guidance source (*e.g.*, as a query in the cross-attention mechanism). However, this simplistic guidance strategy may not fully capture the intricate relationship between neural activity and behavior. Future research should investigate advanced integration methods to better utilize the complex interactions between these data types, potentially improving the model's ability to represent subtle neural patterns. Moreover, while the current study focuses on temporal behavioral signals that correspond directly with neural activity at each time step, future research should aim to extend BLEND to more generalized settings, as we initially explored in **Appendix** A.10. This expansion could encompass non-temporal or discrete behavioral signals, thereby broadening the framework's applicability to diverse neuroscientific domains. Such extensions could prove valuable in investigating sleep stages, categorizing social behaviors, and exploring other scenarios where behavior is not continuously paired with neural activity. This generalization would significantly enhance the versatility and utility of BLEND across a wider spectrum of neuroscience research paradigms.

## ACKNOWLEDGMENTS

This work was supported by the Hong Kong Innovation and Technology Commission (Project No. GHP/006/22GD and ITCPD/17-9), HKUST (Project No. FS111), and the Research Grants Council of the Hong Kong Special Administrative Region, China (Project No. T45-401/22-N).

## REPRODUCIBILITY STATEMENT

To enhance the reproducibility of this study, we provide an Appendix section comprising six sub-sections that offer detailed supplementary information. Appendix A.1 presents the pseudo-code of our proposed BLEND framework, followed by a comprehensive explanation of the behavior-guided knowledge distillation strategies. Additionally, Appendix A.2 provides detailed descriptions of the datasets utilized, namely MC-Maze, MC-RTT, Area2-Bump, and the multi-modal neural activity datasets. To facilitate a deeper understanding of the tasks conducted in this study, Appendix A.3 elucidates the specifics of neural activity prediction, behavior decoding, and match to PSTHs tasks. Subsequently, Appendix A.4 delineates the model configurations, including architecture and hyper-parameters. To further elucidate the superior performance of our proposed method, Appendices A.5 and A.6 offer additional visualizations comparing different base models for distillation and various distillation strategies, enabling qualitative assessment. Our source code will be made publicly accessible upon acceptance of this paper.

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

# A APPENDIX

## A.1 SUPPLEMENTARY CONTENTS OF BLEND ALGORITHM

### A.1.1 BLEND ALGORITHM

---

**Algorithm 1** Behavior-guided Teacher-Student Knowledge Distillation Framework (BLEND)

---

**Require:** Training data $\mathcal{D} = \{(\mathbf{x}_i, \mathbf{b}_i)\}_{i=1}^{N}$, Teacher model $f_{\theta_T}$, Student model $f_{\theta_S}$
**Ensure:** Trained student model $f_{\theta_S}$
 1: // Train teacher model
 2: Initialize teacher model parameters $\theta_T$
 3: **for** each epoch **do**
 4:     **for** each batch $(\mathbf{x}_b, \mathbf{b}_b)$ in $\mathcal{D}$ **do**
 5:         Generate random mask $\mathbf{m}_b \sim p(\mathbf{m})$
 6:         Apply the generated mask $\mathbf{m}_b$ to original input $\mathbf{x}_b$
 7:         Create unmasked portions $\mathbf{x}_{\bar{m}_b}$
 8:         $\hat{\mathbf{x}}_b \leftarrow f_{\theta_T}(\mathbf{x}_{\bar{m}_b}, \mathbf{b}_b)$                    ▷ Teacher's predictions
 9:         Compute $\mathcal{L}_{\text{MTM}}$ using Eq. (1)
10:         Update $\theta_T$ using $\nabla \mathcal{L}_{\text{MTM}}$
11:     **end for**
12: **end for**
13: // Train student model
14: Initialize student model parameters $\theta_S$
15: **for** each epoch **do**
16:     **for** each batch $(\mathbf{x}_b, \mathbf{b}_b)$ in $\mathcal{D}$ **do**
17:         Generate random mask $\mathbf{m}_b \sim p(\mathbf{m})$
18:         Apply the generated mask $\mathbf{m}_b$ to original input $\mathbf{x}_b$
19:         Create unmasked portions $\mathbf{x}_{\bar{m}_b}$
20:         $\mathbf{z}_T \leftarrow f_{\theta_T}(\mathbf{x}_{\bar{m}_b}, \mathbf{b}_b)$                    ▷ Teacher's predictions
21:         $\mathbf{z}_S \leftarrow f_{\theta_S}(\mathbf{x}_{\bar{m}_b})$                       ▷ Student's predictions
22:         Compute $\mathcal{L}_{\text{distill}}$ using Eq. (2):
23:         $\mathcal{L}_{\text{KD}} \leftarrow \alpha \mathcal{L}_{\text{MTM}}(\mathbf{z}_S, \mathbf{x}_b) + (1 - \alpha)\mathcal{L}_{\text{distill}}(\mathbf{z}_S, \mathbf{z}_T)$
                                    ▷ Distillation loss $\mathcal{L}_{\text{distill}}$ chosen from one of Eq. (3), (4), (5), or (6)
24:         Update $\theta_S$ using $\nabla \mathcal{L}_{\text{KD}}$
25:     **end for**
26: **end for**
27: **return** $f_{\theta_S}$

---

### A.1.2 DISTILLATION STRATEGIES.

**Correlation-Distill.** As shown in Section 3.2.1, this distillation strategy aims to preserve the correlation structure of the teacher's outputs. Concretely, the correlation matrix is computed as:

$$\text{Corr}(\mathbf{Y}) = \frac{(\mathbf{Y} - \bar{\mathbf{Y}})(\mathbf{Y} - \bar{\mathbf{Y}})^T}{\sqrt{\text{diag}((\mathbf{Y} - \bar{\mathbf{Y}})(\mathbf{Y} - \bar{\mathbf{Y}})^T) \cdot \text{diag}((\mathbf{Y} - \bar{\mathbf{Y}})(\mathbf{Y} - \bar{\mathbf{Y}})^T)^T}} \tag{7}$$

Here, $\mathbf{Y} \in \mathbb{R}^{N \times T}$ represents the output of either the student or teacher model for a single sample, with $N$ being the number of neurons and $T$ the number of time steps. $\bar{\mathbf{Y}}$ is the mean of $\mathbf{Y}$ along the time dimension, and $\text{diag}(\cdot)$ extracts the diagonal of a matrix. The resulting correlation matrix has dimensions $N \times N$. Note that we use the Frobenius norm $\|\cdot\|_F$ to compute the difference between correlation matrices, as it provides a scalar measure of the overall difference between these matrices.

## A.2 SUPPLEMENTARY CONTENTS OF DATASETS INCLUDED

**MC-Maze** dataset (Churchland et al., 2010; 2012) comprises recordings from the primary motor and dorsal premotor cortices of a monkey performing instructed-delay reaching tasks in 108 different configurations, involving various target positions and virtual barriers. This dataset includes one full session with 2,869 trials and 182 neurons, along with hand kinematics. This dataset is notable for

its behavioral richness, stereotyped repetitions, high trial counts, and clean separation of preparation and execution phases, making it valuable for studying neural population dynamics during movement.

**MC-RTT** dataset (O'Doherty et al., 2017; Makin et al., 2018) features motor cortical recordings during a random target task, comprising continuous, point-to-point reaches with variable lengths and locations, without delay periods. It spans 15 minutes of continuous reaching, artificially divided into 1,351 600ms trials, and includes data from 130 neurons along with simultaneous hand kinematics. Unlike the MC-Maze dataset, MC-RTT introduces modeling challenges due to its non-stereotyped movements, lack of trial repetitions, and the unpredictability of random data snippets, precluding simple trial-averaging approaches for de-noising. This dataset tests models' ability to infer latent representations from single-trial data and detect unpredictable inputs to population activity, offering a more naturalistic benchmark for latent variable models.

**Area2-Bump** dataset (Chowdhury et al., 2020) contains neural recordings from area 2 of the somatosensory cortex, which processes proprioceptive information, as a monkey performed a visually-guided reaching task using a manipulandum. It comprises 462 trials with data from 65 neurons, along with hand kinematics and perturbation information, where in 50% of random trials, the monkey's arm was unexpectedly bumped before the reach cue. This dataset challenges models to infer inputs describing activity after sensory perturbations and to perform robustly with low neuron counts, offering insights into the distinct dynamics of somatosensory areas compared to motor areas.

For each of the above datasets, neural activity recordings are counted in 5ms bins, and behavior signals are also measured at 5ms intervals. Other preprocessing information can be found in Pei et al. (2021)[2] or Ye & Pandarinath (2021)[3].

**Multi-model neural activity dataset** collected by Bugeon et al. (2022) combines neural recordings, genetic information, and behavioral data from mice. This comprehensive dataset includes calcium imaging from the primary visual cortex of four mice (SB025, SB026, SB028, SB030), encompassing over 9,700 neurons across 17 recording sessions. Each session captures roughly 500 neurons for about 20 minutes at a sampling rate of 4.3 Hz. The genetic component comprises expression data for 72 specific genes, enabling the classification of neurons into excitatory or inhibitory types, with further subtyping available for a portion of inhibitory neurons in one mouse (SB025). Complementing the neural data are behavioral recordings such as running speed and pupil dilation, as well as overall brain state classifications. The dataset also provides spatial information for the recorded neurons, offering a rich resource for investigating the interplay between neural activity, cell types, and behavior.

For the preprocessing details of this dataset, please refer to Mi et al. (2023).[4]

### A.3 SUPPLEMENTARY CONTENTS OF TASKS INCLUDED

**Neural activity prediction**. This task requires the model to predict neural activity for held-out neurons. *Co-bps*, also named *Co-smoothing*, is the primary evaluation metric used in the NLB'21 to assess how well models can predict held-out neural activity. The test data is split into held-in and held-out neurons, with models using the training data and held-in test neurons to predict firing rates $\lambda$ for the held-out test neurons. Performance is measured using log-likelihood under a Poisson observation model:

$$p(\hat{\mathbf{y}}_{n,t}) = \text{Poisson}(\hat{\mathbf{y}}_{n,t}; \lambda_{n,t}),$$

where $\hat{\mathbf{y}}_{n,t}$ is the held-out spike count for neuron $n$ at time $t$. The log-likelihood $\mathcal{L}(\lambda; \hat{\mathbf{y}})$ is summed over all held-out neurons and time points, then normalized to "bits per spike" by comparing to a baseline model that uses only the mean firing rate of each neuron:

$$\text{bits/spike} = \frac{1}{n_{sp} \log 2} (\mathcal{L}(\lambda; \hat{\mathbf{y}}_{n,t}) - \mathcal{L}(\bar{\lambda}_n; \hat{\mathbf{y}}_{n,t})),$$

where $\bar{\lambda}_{n,:}$ is the mean firing rate for neuron n and $n_{sp}$ is its total number of spikes. A positive bits per spike (bps) score indicates the model predicts time-varying activity better than the mean firing rate baseline. The co-smoothing metric allows standardized comparison across different types of

---

[2] https://github.com/neurallatents/nlb_tools/tree/main
[3] https://github.com/snel-repo/neural-data-transformers
[4] https://github.com/lumimim/NeuPRINT

models and neural datasets, as it only requires models to output predicted firing rates rather than imposing constraints on model architecture or training (Pei et al., 2021).

**Behavior decoding**. As mentioned in Section 4.1, following the settings in Pei et al. (2021); Ye & Pandarinath (2021), we evaluate the performance of behavior decoding by fitting a ridge regression model. This linear mapping is enforced for all models in the decoding process to prevent complex decoders from compensating for poor neural dynamics estimation. More sophisticated decoders could potentially achieve better performance but at the cost of obscuring the quality of the underlying latent representations. The behavioral data used for decoding in MC-Maze, MC-RTT, and Area2-Bump is monkey hand velocity.

**Match to PSTHs**. This task evaluates how well models can reproduce the stereotyped neural responses captured by peri-stimulus time histograms (PSTHs). PSTHs are computed by averaging neuronal responses across trials within a given condition, revealing consistent features of neural activity. For datasets with clear trial structures (MC-Maze and Area2-Bump), the task computes the $R^2$ between trial-averaged model rate predictions for each condition and the true PSTHs, first for each neuron across all conditions and then averaged across neurons.

## A.4 Supplementary contents of model configurations

**LFADS (Latent Factor Analysis via Dynamical Systems)** (Pandarinath et al., 2018) is a deep learning method designed to model neural population dynamics from single-trial spiking data, which utilizes recurrent neural networks (RNNs) to model the underlying dynamics of neural populations. The model consists of an encoder RNN that compresses the input spike data into a latent code, and a generator RNN that reconstructs the data from this latent representation. LFADS infers low-dimensional latent factors and initial conditions for each trial, which are then used to generate de-noised estimates of neural firing rates. This architecture allows LFADS to capture complex, non-linear dynamics in neural data while providing interpretable latent representations of neural population activity on single trials.

As the base model for our privileged knowledge distillation with behavior information in neural dynamics modeling, we set the hidden dimension to 64 and factor size to 32. For model training, we set the batch size to 64, learning rate to $1 \times 10^{-3}$ with 5000 warm-up iterations and weight decay to $5 \times 10^{-5}$. Mask ratio is set to 0.25.

**NDT (NeuralDataTransformer)** (Ye & Pandarinath, 2021) is a non-recurrent architecture designed to model neural population spiking activity, based on the BERT encoder. The core of the NDT consists of a stack of Transformer layers (typically 6 layers), each containing self-attention mechanisms, layer normalization, and feedforward neural networks. It uses masked modeling during training, where random portions of the input sequence are masked and the model is trained to reconstruct the original input, encouraging it to leverage contextual information. The NDT processes neural data in parallel rather than sequentially, enabling faster inference times compared to recurrent models like LFADS, while achieving comparable performance in modeling autonomous neural dynamics and enabling accurate behavioral decoding.

As the base model for our behavior-guided knowledge distillation in neural dynamics modeling, we set the Transformer layer number to 4 for MC-Maze, MC-RTT, and Area2-Bump datasets, hidden dimension to 128, and number of attention heads to 2. For model training, we set the batch size to 64, learning rate to $1 \times 10^{-3}$ with 5000 warm-up iterations and weight decay to $5 \times 10^{-5}$. Mask ratio is set to 0.25.

**NeuPRINT** (Mi et al., 2023) is designed to assign time-invariant representations to individual neurons based on population recordings of neural activity. The model uses a transformer architecture to implement an implicit dynamical system that predicts neural activity based on the past activity of the neuron itself and permutation-invariant statistics of the population activity. NeuPRINT learns both the dynamical model and time-invariant representations for each neuron by minimizing the prediction error of future neural activity. The learned representations can then be used for downstream tasks such as predicting transcriptomic cell types.

As the base model for our behavior-guided knowledge distillation in transcriptomic identity prediction, we follow the same configurations in Mi et al. (2023) and set the dimension of time-invariant

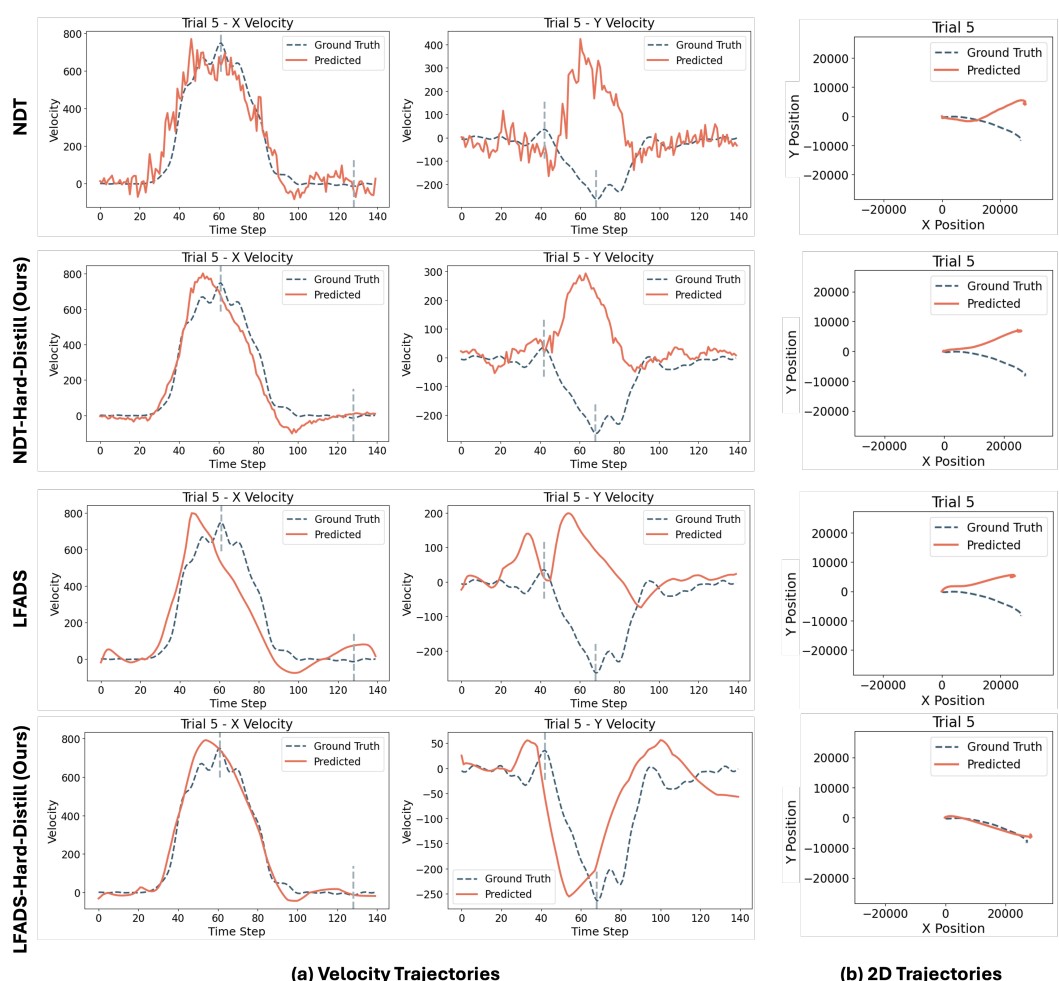

Figure 4: Visualization of predicted hand velocity on MC-Maze dataset of base model NDT and LFADS, as well as their behavior-guided distilled counterparts. (a) Prediction and ground truth of X and Y velocities, respectively. (b) Prediction and ground truth of 2D hand movement trajectories.

embedding to $64$. For model architecture, $1$ transformer layer with $2$ attention heads is used. For model training, the batch size is set to $1024$ and the learning rate is set to $1 \times 10^{-3}$.

### A.5 Additional qualitative analysis on NDT and LFADS

This section provides more qualitative analysis based on visualization of model predictions in the behavior decoding task. As shown in Fig. 4, a comprehensive visualization comparing the performance of base models (NDT and LFADS) with their behavior-guided distilled versions (NDT-Hard-Distill and LFADS-Hard-Distill) on the MC-Maze dataset is presented.

#### A.5.1 1D Hand Velocity Decoding (X and Y axes)

Fig. 4(a) shows the decoded X and Y hand velocities compared to the ground truths.

**NDT vs. NDT-Hard-Distill:** The distilled NDT model demonstrates significant improvements in capturing both X and Y velocity components simultaneously. The base NDT model shows considerable deviations from the ground truth, particularly in the latter half of the time series. In contrast, NDT-Hard-Distill tracks both velocity components much more accurately throughout the entire se-

quence. It better captures the magnitude and timing of velocity changes in both dimensions, resulting in a much closer match to the ground truth trajectory.

**LFADS vs. LFADS-Hard-Distill:** Both LFADS and the distilled model perform well in tracking the X and Y velocities, but the distilled version shows noticeable improvements. LFADS-Hard-Distill more accurately captures the peaks and troughs of both velocity components, especially in the middle and latter parts of the sequence. The refinements are subtle but consistent, indicating a better overall representation of the hand movement dynamics. Notably, in the trough region of the Y velocity component, NDT, NDT-Hard-Distill, and LFADS exhibit a significant mischaracterization of the velocity profile, predicting a peak where the ground truth demonstrates a trough. In contrast, LFADS-Hard-Distill accurately captures this critical feature, successfully predicting the trough trend in concordance with the ground truth.

### A.5.2 2D Hand Movement Trajectory Decoding

Fig. 4(b) presents a comparison of 2D hand movement trajectories for four models: NDT (base model), NDT-Hard-Distill (our distilled model), LFADS (base model), and LFADS-Hard-Distill (our distilled model), alongside the ground truth trajectory.

For NDT, NDT-Hard-Distill, and LFADS, there's a significant discrepancy between the ground truth and predicted trajectories. While the ground truth curves downward, the predictions curve upward, indicating poor performance or a fundamental misunderstanding of the underlying pattern. For LFADS-Hard-Distill, the predicted trajectory closely follows the ground truth, suggesting much better performance for this model. Moreover, LFADS-Hard-Distill appears to have successfully learned the system's behavior, demonstrating superior predictive capabilities.

### A.5.3 Comparative Analysis

Overall, we can conclude that the distillation process benefits both NDT and LFADS, with more pronounced improvements observed in the NDT model. The distillation process enhances the models' ability to capture fine-grained details and reduce erratic predictions, resulting in smoother, more accurate velocity profiles. Meanwhile, LFADS-Hard-Distill demonstrates superior performance in accurately predicting the trough region, as illustrated in Fig. 4(a), and in capturing the trajectory trend, as shown in Fig. 4(b). In both instances, the other three models exhibit significant deviations from the ground truth. This qualitative observation aligns with the quantitative results presented in Tab. 3, wherein LFADS-Hard-Distill consistently outperforms the other three models in behavior decoding.

These visualizations provide strong evidence for the effectiveness of the BLEND framework in improving neural dynamics modeling. Moreover, the distillation process appears to transfer valuable information from the behavior-guided teacher model to the student model, enhancing its ability to infer accurate hand velocities in multiple dimensions from neural activity alone. The improvements are consistent across different aspects of the movement (2D movement trajectories, 1D X and Y velocities), suggesting a comprehensive enhancement of the models' predictive capabilities.

In summary, this analysis demonstrates that the BLEND framework not only improves quantitative metrics but also leads to qualitatively better predictions of complex behavioral outputs from neural activity. These visualizations provide intuitive and compelling evidence for the benefits of incorporating behavioral information through privileged knowledge distillation in neural dynamics modeling.

### A.6 Additional qualitative analysis on different distillation strategies

This section presents the qualitative analysis of different distillation strategies we employ in this work, including Hard Distillation (Eq. 3), Soft Distillation (Eq. 4), Feature Distillation (Eq. 5), and Correlation Distillation (Eq. 6). This analysis is conducted based on the visualization of model predictions in the behavior decoding task of the MC-Maze dataset. As illustrated in Fig. 5, we compare the performance of NDT and its different behavior-guided distilled variants in decoding hand movement trajectories and velocities.

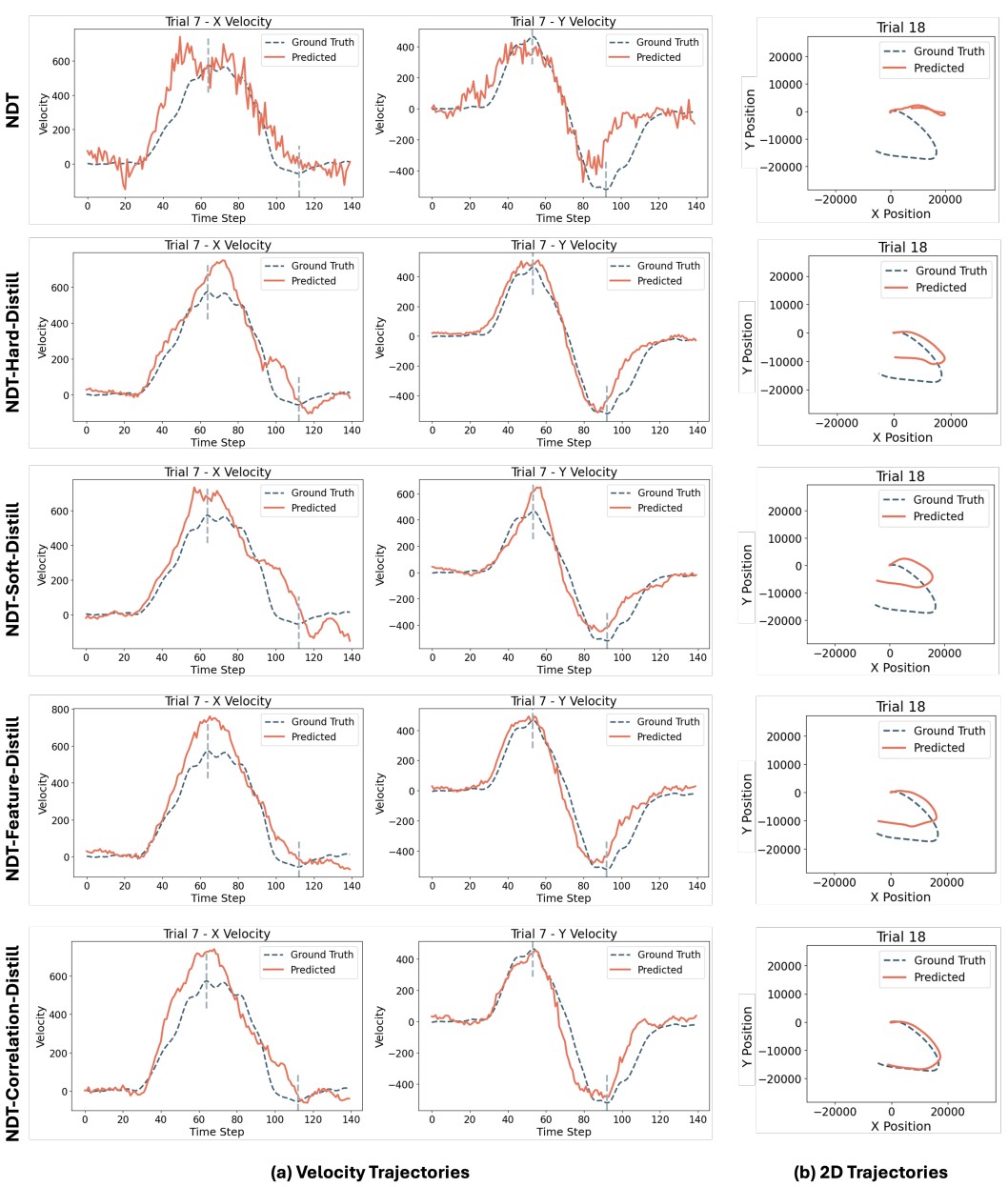

Figure 5: Visualization of predicted hand velocity on MC-Maze dataset of different behavior-guided distilled models, including based model NDT, NDT-Hard-Distill, NDT-Soft-Distill, NDT-Feature-Distill, and NDT-Correlation-Distill. (a) Prediction and ground truth of X and Y velocities, respectively. (b) Prediction and ground truth of 2D hand movement trajectory.

### A.6.1    1D Hand Velocity Decoding (X and Y axes)

For X velocities, all distilled models show improved alignment with ground truth compared to the base model NDT, especially during peak velocity periods (time steps 40-80). For the Y velocities, our behavior-guided distilled models demonstrate better tracking of the ground truth, with reduced oscillations and improved accuracy in predicting direction changes. Moreover, the distillation approaches are particularly effective in capturing the synchronous changes in X and Y velocities, suggesting better prediction of overall motion dynamics. NDT-Correlation-Distill stands out with its ability to accurately predict velocities in both dimensions simultaneously, indicating a strong grasp of the interdependencies between X and Y motions.

### A.6.2    2D Hand Movement Trajectory Decoding

For 2-dimensional hand movement trajectory decoding, the base model NDT shows significant deviation from the ground truth, especially in the lower part of the trajectory. In contrast, all distillation methods demonstrate improved trajectory prediction, with closer alignment to the ground truth path. Specifically, NDT-Correlation-Distill appears to provide the most accurate trajectory prediction, closely matching the ground truth's shape and endpoints.

### A.6.3    Model-specific Observations

For each distillation strategy employed in this study, the following key observations can be discerned from Figure 5:

- NDT-Hard-Distill: Shows substantial improvement over base NDT in both velocity and trajectory prediction.
- NDT-Soft-Distill: Offers improved performance, though with some oscillations in velocity predictions.
- NDT-Feature-Distill: Demonstrates good overall performance, with smooth velocity curves and accurate trajectory prediction.
- NDT-Correlation-Distill: Appears to be the best-performing method, showing excellent alignment with ground truth across all metrics.

Overall, we could conclude that the behavior-guided distillation approaches clearly enhance the predictive capabilities of the NDT model. The improvements are particularly noticeable in handling complex motions and maintaining accuracy over longer time horizons. The correlation-based distillation method seems to be the most effective, suggesting that preserving relational information during distillation is crucial for accurate predictions.

### A.7    Cross-correlation Analysis

This subsection provides cross-correlation and mutual information analyses on the MC-Maze dataset to quantify the relationship between neural activity and behavioral variables.

The cross-correlation heatmap was generated by computing correlations between each neuron's activity and behavioral variables (X and Y velocities) across different time lags. For each neuron, the data was flattened across trials and timepoints, detrended, and normalized before computing correlations using a Fast Fourier Transform (FFT) method. The resulting correlations were arranged into a matrix where each row represents a neuron, each column represents a time lag, and the color intensity indicates the correlation strength. As shown in Fig. 6, our analysis revealed robust temporal relationships between neural activity and movement kinematics. Nearly half of the recorded neurons ($47.4\%$) exhibited leading relationships with horizontal (X) velocity, while a slightly larger proportion ($54.7\%$) led vertical (Y) velocity components. Importantly, we observe median leads of 20ms (X) and 10ms (Y), indicating that neural activity and behavior exhibit complex, complementary dynamics.

The strength of these neural-behavioral relationships was confirmed by highly significant correlations for both movement components ($p < 0.001$ for both X and Y velocities). This robust statistical coupling explains why behavior serves as an effective privileged information source in our BLEND

knowledge distillation framework, *i.e.*, it provides a reliable signal about the neural computations being performed. Crucially, we find that population-level mutual information (MI) is substantially higher than single-neuron MI, indicating that behavior captures coordinated neural dynamics that emerge at the population level. This difference demonstrates that behavior provides a window into population-level neural representations, which is a key motivation for our knowledge distillation approach.

Taken together, these analyses show that neural activity and behavior in the MC-Maze dataset are strongly coupled, with behavior offering complementary, contextual information about the underlying neural computations. By effectively leveraging these neural-behavioral relationships, our BLEND framework is able to achieve significant performance improvements in learning better neural population representations. The robust temporal structure and statistical significance of the observed cross-correlations further validate the importance of behavior as a privileged signal in our knowledge distillation approach.

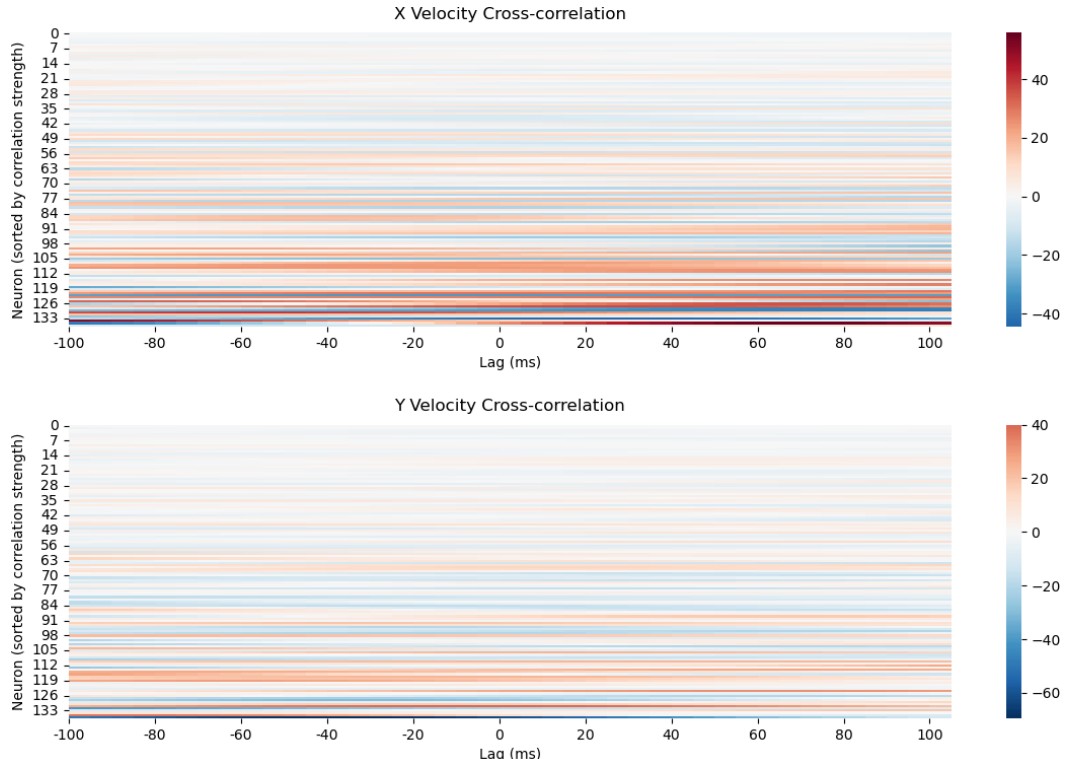

Figure 6: Visualization of neural activity and behavior cross-correlation heatmap on MC-Maze dataset. Upper: cross-correlation between independent neurons and horizontal movement; Below: cross-correlation between independent neurons and vertical movement.

## A.8 EMPIRICAL STUDY ON BEHAVIOR GUIDANCE

To understand why incorporating behavior signals leads to better neural activity reconstruction, we conducted detailed analyses of the relationships between behavioral and neural data (exemplified with MC-Maze dataset).

### A.8.1 NEURAL SPACE ORGANIZATION

We first examined how behavior signals influence the organization of neural representations using t-SNE visualization (Figure 7). The baseline model's representations (shown in red) appear as scattered, disconnected clusters across the neural space. In contrast, BLEND's representations (shown in blue) form more cohesive structures, suggesting that behavior signals help constrain neural activ-

ity patterns into meaningful manifolds. This improved organization likely contributes to BLEND's superior reconstruction capability.

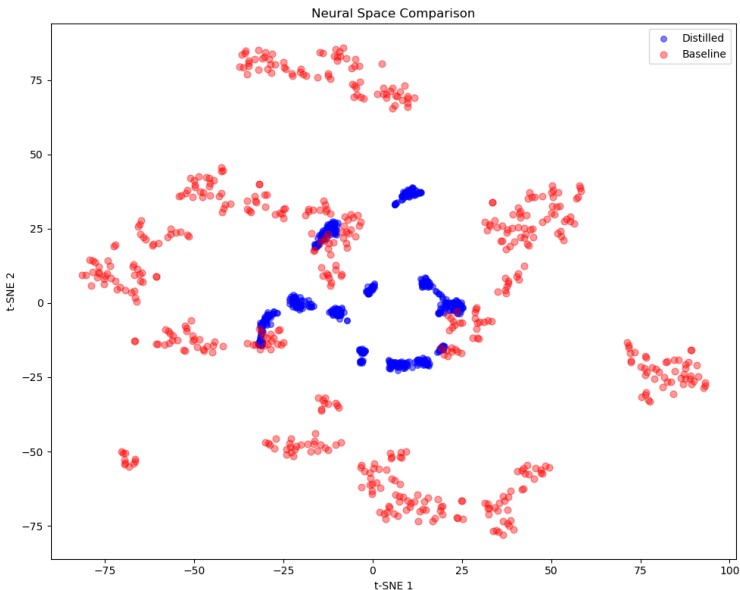

Figure 7: t-SNE visualization of neural representations showing the organization of neural space. Red points represent baseline model representations, while blue points show BLEND model representations. The more cohesive clustering of BLEND's representations suggests better capture of underlying neural structure.

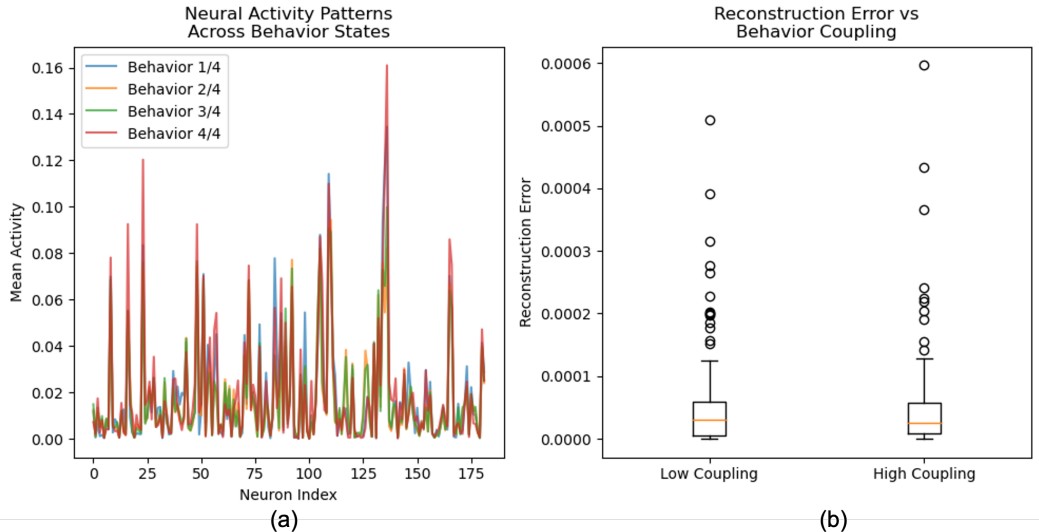

Figure 8: Behavior-neural analysis. (a) Neural activity patterns across different behavioral states show systematic variation. Different colors represent different behavior quartiles. (b) Box plots comparing reconstruction errors for neurons with low versus high behavior coupling, demonstrating improved reconstruction for behavior-coupled neurons.

### A.8.2 BEHAVIOR-DEPENDENT NEURAL ACTIVITY PATTERNS

We analyzed how neural activity patterns vary systematically with behavioral states (Figure 8, left). The behavioral states were determined through the behavior data, which consists of two continu-

ous variables (x and y coordinates of hand position) over time. We used the first behavior variable (x coordinate) and divided it into quartiles using numpy's percentile function: behavior_bins = percentile(behavior_data, [0, 25, 50, 75, 100]). This created four behavior states (1/4 through 4/4), representing different ranges of hand positions. For each state, we computed the mean neural activity pattern across all time points falling within that state. The resulting patterns show clear modulation of neural activity by behavioral state, with distinct activity profiles for different hand positions. Some neurons (e.g., indices 25, 50, and 125) show particularly strong behavior-dependent modulation.

### A.8.3 RECONSTRUCTION QUALITY ANALYSIS

We quantitatively assessed how behavior coupling influences reconstruction quality through a comprehensive analysis framework. First, we quantified behavior-neural coupling using mutual information (MI): $MI(N_i, B) = \sum_{n,b} p(n,b) \log \frac{p(n,b)}{p(n)p(b)}$, where $N_i$ is the activity of neuron $i$ and $B$ is the behavior variable. We computed this for each neuron using scikit-learn's mutual_info_regression function. Neurons were then classified into high coupling (MI > median) and low coupling (MI $\leq$ median) groups.

For each neuron $i$, we computed the reconstruction error as $E_i = \frac{1}{T} \sum_{t=1}^{T} (\hat{r}_i^t - r_i^t)^2$, where $\hat{r}_i^t$ is the predicted firing rate and $r_i^t$ is the true firing rate at time $t$. The box plots (Figure 8, right) show the distribution of these errors for both groups. Statistical analysis revealed significant differences between high and low coupling groups (Wilcoxon rank-sum test: $p < 0.001$), with high-coupling neurons showing consistently lower reconstruction errors.

These analyses collectively demonstrate that BLEND's improved performance stems from its ability to leverage inherent structure in behavior-neural relationships. The systematic variation of neural activity with behavior and the improved reconstruction for behavior-coupled neurons provide strong empirical evidence for the effectiveness of incorporating behavior signals in neural activity reconstruction.

### A.9 DISTILLATION STRATEGY EVALUATION ON SYNTHETIC DATA

This subsection provides a detailed exploration and a comprehensive understanding of when and why different strategies excel. We conduct additional experiments using carefully designed synthetic datasets that mirror the structure of real neural recordings while allowing precise control over neural-behavioral relationships.

We created three types of synthetic datasets with distinct characteristics.

- `Simple`: The first type implements simple linear relationships between neural activity and behavior, designed to test basic knowledge transfer. In this dataset, neural activity is directly derived from sinusoidal behavioral trajectories through a linear transformation. This creates a clear and interpretable relationship between behavior and neural responses, with added Poisson noise to simulate realistic spike counts.

- `Hierarchical`: The second type features hierarchical relationships where different neuron groups encode behaviors at varying timescales, simulating the layered processing often observed in neural circuits. The hierarchical dataset introduces temporal complexity by dividing neurons into three functional groups, each processing behavior at different timescales. Fast neurons respond to immediate behavior, medium neurons integrate over 5 timepoints, and slow neurons average over 10 timepoints, creating a rich temporal hierarchy that mirrors the multi-timescale processing observed in real neural systems.

- `Complex`: The third type incorporates complex population-level correlations with temporal dependencies, mimicking the intricate dynamics found in real neural populations. The complex population dataset represents the most sophisticated model, organizing neurons into five correlated assemblies with explicit population-level structure. This dataset features decaying sinusoidal behavior patterns and maintains specific correlation structures (0.3 within groups) through multivariate normal noise, creating realistic population dynamics.

| Methods | Simple | Hierarchical | Complex |
|---------|--------|--------------|---------|
| LFADS-Hard-Distill | 0.985 | 0.976 | 0.182 |
| NDT-Hard-Distill | 0.934 | 0.926 | 0.147 |
| NDT-Soft-Distill | 0.927 | 0.932 | 0.146 |
| NDT-Feature-Distill | 0.926 | 0.940 | 0.139 |
| NDT-Correlation-Distill | 0.935 | 0.944 | 0.180 |

Table 4: Results showing variants of BLEND framework on three synthetic datasets with distinct characteristics. Vel-$R^2$ is used as the measurement metric.

All three datasets share common dimensions (2,869 trials, 140 timepoints, 182 neurons, 2 behavioral variables) and are systematically divided into train/eval splits and held-in/held-out neurons, enabling rigorous testing of neural analysis methods.

Our experiments with these synthetic datasets revealed clear patterns that explain the performance differences observed in Table 4. LFADS consistently performed better with Hard Distillation across all synthetic cases, particularly excelling with simple linear relationships (around 6% improvement over NDT-based variants in Vel-$R^2$). This aligns with our understanding that LFADS's RNN-based architecture, with its inherent temporal continuity and built-in constraints, benefits most from direct knowledge transfer. The architecture's strong internal regularization makes it well-suited for precise, deterministic knowledge transfer through hard distillation.

Conversely, NDT showed superior performance with Correlation Distillation, especially in cases with complex population-level correlations (around 20% improvement over other NDT-based variants). This finding stems from NDT's transformer-based architecture, which processes information in parallel and lacks inherent temporal constraints. The correlation distillation strategy provides valuable structural guidance that complements NDT's flexible architecture, helping it maintain important population-level relationships in the learned representations.

**Intuitions and guidelines:** These insights from synthetic data experiments provide concrete guidelines for strategy selection in practice. For architectures with strong internal regularization like LFADS, Hard Distillation offers the most direct and effective knowledge transfer. For more flexible architectures like NDT, Correlation Distillation helps maintain complex population dynamics and temporal relationships. Soft Distillation, serving as a robust middle-ground option, can be beneficial for datasets where subtle variations in neural responses need to be preserved, though it may not fully capture complex population-level dynamics. Feature Distillation, while generally showing more modest improvements, is particularly effective when the neural data exhibits a clear hierarchical structure, as it can capture relationships across different levels of neural representation. This understanding not only explains the performance patterns in our original results but also offers a principled approach to selecting distillation strategies for future applications of BLEND to different neural datasets and architectures.

## A.10 Extending BLEND to Discrete or Non-temporal Behavior Labels

To verify the capability of BLEND with discrete or non-temporal behavior observations, we extend BLEND on the DMFC-RGS dataset from NLB'21 Benchmark Pei et al. (2021), which contains recordings from dorso-medial frontal cortex (DMFC) during a cognitive timing task known as Ready-Set-Go (RSG). The behavioral variables in this task are distinctly discrete in nature. The task incorporates five key behavioral features, represented as binary or categorical variables: 'is_eye' indicating whether the response was an eye movement (1) or joystick movement (0), 'theta' specifying the categorical direction of movement (left or right), and 'is_short' denoting whether the trial used the Short prior (1) or Long prior (0). The remaining variables 'ts' and 'tp' represent the sample and produced time intervals respectively. These discrete behavioral choices create a structured experimental design with clearly defined conditions.

Given that this dataset has discrete and non-temporal behavior labels, we align the neural activity and behavior by introducing a learnable temporal position encoding for discrete behavior variables across the length of neural recording. We compare NDT-Hard-Distill with the original NDT model on this dataset, and this simple extension shows improvement in both Co-Bps (0.127 achieved by

NDT-Hard-Distill and $0.118$ achieved by NDT) and PSTH-$R^2$ ($0.444$ achieved by NDT-Hard-Distill and $0.338$ achieved by NDT).

These promising results demonstrate the effectiveness of BLEND on both continuous/discrete and temporal/non-temporal behavior observations, showcasing a flexible solution we provide for learning better neural population dynamics.

