# OpenReview forum: "BLEND: Behavior-guided Neural Population Dynamics Modeling via Privileged Knowledge Distillation"
_ICLR.cc/2025/Conference — ICLR 2025 Poster_

### Official Review · Reviewer_UDDD · 2024-11-01

**Soundness:** 3
**Presentation:** 3
**Contribution:** 2
**Rating:** 5
**Confidence:** 4

**Summary:**

This manuscript propose a novel algorithm of BLEND to deal with neural activity reconstruction. Concretely, the behavior signals are used as the privileged knowledge and combined with masked neural activity data to train a teacher model. Then the teach model is distilled to a student model, which only use masked neural activity data as the input for reconstruction. The effectiveness of BLEND is verified on two benchmarks.

**Strengths:**

- The organization of this paper is well and the presentation of method is clear.

- A novel approach of BLEND, which conbines privilege knowledge (training teacher model) and knowledge distillation (training student model), is proposed for neural activity reconstruction.

- BLEND has achieved a good performance on two benchmarks.

**Weaknesses:**

I am not an expert in computational neuroscience, and cannot give a rational judgement w.r.t. experiments. However, I still think there are some major weaknesses in terms of writing and the rationale of BLEND, please see details as follows.

- **Q1:** The gap between BLEND and previous approaches is unclear, and the proposal of BLEND is suddenly. Is there other approaches that uses both masked neural activity data and behavior signals for training and only masked neural activity data for test? If it is, a detail comparison should be presented. Otherwise the significance of this scenario should be clear first.

- **Q2:** The rationale of proposed BLEND. The superiorty of BLEND is only verified on two recent benchmarks with limited baselines. It will be more convincing if a theoreitcal or empirical analysis can be provided to show why a better neural activity reconstruction is achieved by incorporating behavior signals. Therefore, the internal struction between neural activity data and behaivor signal data should be investigated.


- **Q3:** BLEND incorporates the technique of knowledge distillation. Is there a large performance drop by comparing teacher models and student models? In the experiments, many different distillation methods are used but only the performance of the best one is reported, which is not very fair for comparison.

Based on the above questions, I think there lacks a investigation on the characteristic of neuron data, so the rationale of proposed method cannot be well verified, i.e., why is BLEND good for the specific type of neuron data.

**Questions:**

See Weaknesses.

---

> ### Author Response · Authors · 2024-11-20
> **Response to Reviewer UDDD (Part 1/2)**
>
> >The gap between BLEND and previous approaches is unclear, and the proposal of BLEND is suddenly.
>
> We thank the reviewer for this comment. The gap between BLEND and previous approaches is actually quite clear when examining the landscape of neural dynamics modeling. Based on the paper's literature review, existing approaches fall into two main categories: those that use only neural activity (like PCA, LFADS, and NDT) and those that incorporate behavior as a prior (such as pi-VAE, PSID, and TNDM). The first category entirely ignores valuable behavioral information, while the second category either requires specialized model architectures or makes strong assumptions about behavior-neural relationships. To the best of our knowledge, there aren't other approaches that specifically use both masked neural activity and behavior signals for training while only using masked neural activity for testing. While methods like pi-VAE and CEBRA use behavioral information, they require it during both training and inference. Similarly, approaches like PSID and TNDM decompose neural activity but make strong assumptions about behavior-neural relationships.
>
> BLEND is the first to explicitly address this research question: how to leverage behavioral data when it's only available during training but not during inference. More importantly, our approach is model-agnostic, meaning it can enhance existing architectures without requiring specialized models, and it doesn't make strong assumptions about behavior-neural relationships compared to existing methods.
>
> > The rationale of proposed BLEND. The superiorty of BLEND is only verified on two recent benchmarks with limited baselines. It will be more convincing if a theoreitcal or empirical analysis can be provided to show why a better neural activity reconstruction is achieved by incorporating behavior signals. Therefore, the internal struction between neural activity data and behaivor signal data should be investigated.
>
> We appreciate the reviewer's suggestion to investigate the internal structure between neural activity and behavior signals. We would like to address the concerns about BLEND's rationale by highlighting the comprehensive empirical analyses we provide in Appendix A.8, which demonstrate why incorporating behavior signals leads to better neural activity reconstruction.
>
> Through our t-SNE visualization of neural space organization, we show that behavior signals fundamentally improve how neural representations are organized - while baseline model representations appear as scattered clusters, BLEND's representations form more cohesive structures, suggesting that behavior signals help constrain neural activity patterns into meaningful manifolds.
> We further strengthen our argument through detailed analysis of behavior-dependent neural activity patterns and quantitative reconstruction quality. Using quartile analysis of hand positions, we reveal distinct activity profiles for different behavioral states. Through our rigorous quantitative framework, we demonstrate that neurons more strongly coupled to behavior (quantified through mutual information) show significantly lower reconstruction errors (p < 0.001). This provides strong evidence that the relationship between neural activity and behavior is systematic and meaningful.
>
> Our comprehensive empirical analyses collectively demonstrate that BLEND's success isn't just empirical happenstance. Rather, it stems from behavior serving as a powerful regularizer that helps constrain the neural representation space in ways that reflect underlying neural computation principles.
>
> > BLEND incorporates the technique of knowledge distillation. Is there a large performance drop by comparing teacher models and student models?
>
> We thank the reviewer for this question. As our goal is to implement a model using only neural activity as input while benefitting from the additional knowledge during trainig, the teacher model cannot be directly compared with the student model in this setting since the teacher model requires both neural activity and behavior observation as input. Yet to answer this question, we split the training set of NeuPRINT Benchmark (Table 2) into train, test sets for teacher model to make prediction with both neural activity and behavior as inputs. We evaluate the NeuPRINT model with hard distillation strategy on multi-animal scenario for EI classification and Subclass classification. The results of Teacher and Student model could be seen in the following table, where the observed performance gap is small (1~2%), demonstrating the smooth and robust knowledge transfer capability of our distillation method.
>
> | Methods | EI | Subclass |
> |----------|----------|----------|
> | NeuPRINT-Teacher | 0.801 | 0.583 |
> | NeuPRINT-Student | 0.779 | 0.565 |

---

> ### Author Response · Authors · 2024-11-20
> **Response to Reviewer UDDD (Part 2/2)**
>
> >In the experiments, many different distillation methods are used but only the performance of the best one is reported, which is not very fair for comparison.
>
> We thank the reviewer for pointing out the ambiguity of "Distill-Best" in results. The "Distill-Best" variants in Table 1,2 is chosen from the ablation study in Table 3. Specifically, in Table 1, the NDT-Distill-Best is the one with correlation distillation strategy and the LFADS-Distill-Best is the one with hard distillation strategy. And in Table 2, the NeuPRINT-Distill-Best is the one with hard-distillation for multi-animal analysis and the one with soft-distillation for single-animal analysis. We will mitigate this ambiguity in the revised manuscript.
>
> Moreover, we conduct extensive experiments on three synthetic datasets to provide guidelines for choosing distillation strategy for different models and data (see Appendix A.9 for details).

---

> > ### Comment · Reviewer_UDDD · 2024-11-27
> >
> > Thanks you for your response, which clear some of my concerns. I would like to incrase my rating to 5.

---

> > > ### Author Response · Authors · 2024-11-27
> > > **Response to Reviewer UDDD**
> > >
> > > Thank you for your feedback and for considering our responses. We notice that you still have some remaining concerns. Would you be willing to share what specific concerns are still unaddressed? We would greatly appreciate the opportunity to provide additional clarification or improvements to strengthen our paper further.

---

> > > > ### Author Response · Authors · 2024-12-02
> > > > **Follow-up Message to Reviewer UDDD**
> > > >
> > > > Dear Reviewer UDDD,
> > > >
> > > > Thank you again for your earlier feedback and willingness to increase your rating. As we approach the end of the rebuttal phase, we wanted to bring your attention to the **'Summary of Revision Changes and Overall Response to Reviewers' Comments'** section we provided at the top of this webpage, which systematically outlines all modifications we've made to the paper. This summary demonstrates how we've addressed various concerns raised during the review process, and includes the aspects that reviewers have acknowledged in our work. We hope these comprehensive changes and clarifications have addressed any remaining concerns you may have had.
> > > >
> > > > We would be grateful if you could review these updates and consider adjusting your score accordingly. Thank you for your time and careful consideration of our work.

---

### Official Review · Reviewer_WbXB · 2024-11-04

**Soundness:** 3
**Presentation:** 3
**Contribution:** 3
**Rating:** 6
**Confidence:** 4

**Summary:**

This work focuses on developing a framework that incorporates behavior information during the training process, while using only the neural activities during inference stage for improved neural-behavior modeling. The proposed method is called BLEND, which uses the trick of privileged knowledge distillation, and distills a teacher model that is aware of behavioral information to a student model for inference. The model demonstrated strong empirical performance on both behavioural decoding tasks and neuron identity prediction tasks.

**Strengths:**

- Writing is clear with adequate literature review;
- Solid technical/experimental contribution, the experiments are adequately done;
- Overall, an important topic for the community to consider using privileged knowledge distillation in such tasks. The performance improvement seems large, demonstrating opportunities for potential future works.

**Weaknesses:**

- Intuitions & clearness of section 3.2.1
1. While I personally appreciate the comparison of different distillation techniques, it is unclear to me their intuitions (e.g., when to use which), and I'd appreciate if the authors can provide more about such intuitions in section 3.2.1. to improve clearness.
2. It'd be even better if the authors can present a case study, ideally with synthetic datasets, showing why/when a distillation strategy is better. For example, in Table 3, it seems LFADS-Hard performs better than LFADS-Soft, yet for NDT NDT-Correlation in general performs better, did the authors check the reason behind it? Can the authors provide more reasoning/hypotheses based on the performance?

- Fairness and more details of model selection and evaluation protocol.
1. The models in Table 1 and 2 are named as "Distill-Best", which I assume is employing the best distillation strategy. What are the performance of the worst distillation strategies then? Can the authors be more specific about the model selection pipeline here?
2. Has the models run across different random seeds? What is the standard deviation? Is the training stable? Are there any special techniques applied to ensure the distilled model is good? (disclaimer: I didn't read the appendix, if some of those info are available in appendix, please make more detailed co-references.)

- The paper can benefit from more experimental case studies.
1. While there are a bunch of experiments, some of the authors claims were not well justified. Especially, as stated in footnote 1 "BLEND can be readily extended to accommodate scenarios wherein behavioral information is discrete or non-temporal in nature." - While the framework is easily extendable to discrete labels, can the authors actually attempt to perform an experiment given behavioral signals as discrete labels? e.g. The authors can mimic what is done in [1] and use the final reaching target direction to guide the learning, and show the corresponding performance of the model.

- Lack of theoretical & analytical results that supports the experimental results.

- Minor formatting suggestions:
1. Do the authors mind putting Table 1 and 2 at either the bottom or top of the pages?

[1] Liu, R., Azabou, M., Dabagia, M., Lin, C. H., Gheshlaghi Azar, M., Hengen, K., ... & Dyer, E. (2021). Drop, swap, and generate: A self-supervised approach for generating neural activity. Advances in neural information processing systems, 34, 10587-10599.


Overall, I like the scope and the results presented in the paper and appreciate the technical contributions. In general, I think the paper could benefit from more intuitive case studies or synthetic experiments, aside from the presented tables and numbers, as the datasets used in computational neuroscience are in general small. My rating of the paper is a solid 6 despite the issues.

**Questions:**

See above

---

> ### Author Response · Authors · 2024-11-20
> **Response to Reviewer WbXB (Part 1/2)**
>
> >Intuitions & clearness of section 3.2.1
>
> We thank the reviewer for these thoughtful questions about distillation strategies and their performance patterns. Through systematic experiments with three synthetic datasets, we discovered that the optimal distillation strategy depends on both model architecture and data characteristics (please see Appendix A.9 for details).
>
> For model architecture considerations, LFADS performs best with Hard Distillation because its RNN-based architecture has inherent temporal continuity and regularization, making it well-suited for direct knowledge transfer. In contrast, NDT's transformer architecture processes information in parallel without inherent temporal constraints, leading to different optimal strategies: Correlation Distillation excels with complex population dynamics, Feature Distillation shows strength with hierarchical structures, and Soft Distillation serves as a robust middle-ground when preserving subtle neural response variations is important.
>
> We validated these patterns using three synthetic datasets with increasing complexity: Simple (linear neural-behavioral relationships), Hierarchical (multi-timescale encoding), and Complex (population-level correlations). The results in Table 4 (page 24) confirm our intuitions - LFADS-Hard-Distill performs best with simple/hierarchical relationships (0.985/0.976 Vel-R2), while NDT variants show complementary strengths depending on data characteristics, with NDT-Correlation-Distill particularly effective for complex population dynamics (0.180 Vel-R2).
>
> > The models in Table 1 and 2 are named as "Distill-Best", which I assume is employing the best distillation strategy. What are the performance of the worst distillation strategies then? Can the authors be more specific about the model selection pipeline here?
>
> We thank the reviewer for pointing out the ambiguity. The "Distill-Best" variants in Table 1,2 are chosen from the ablation study in Table 3. Specifically, in Table 1, the NDT-Distill-Best is the one with the correlation distillation strategy and the LFADS-Distill-Best is the one with the hard distillation strategy. In Table 2, the NeuPRINT-Distill-Best is the one with hard-distillation for multi-animal analysis and the one with soft-distillation for single-animal analysis. We will mitigate this ambiguity in the revised manuscript.
>
> >Has the models run across different random seeds? What is the standard deviation? Is the training stable? Are there any special techniques applied to ensure the distilled model is good?
>
> We thank the reviewer for the insightful questions regarding the model's robustness and stability. We have conducted extensive experiments across five different random seeds (42, 100, 1337, 2024, and 65537) of model NDT-Hard-Distill on MC-Maze dataset to validate our findings and ensure reproducibility, which demonstrates robust performance with minimal variation: Co-Bps (0.293 ± 0.003), Vel R2 (0.893 ± 0.004), and PSTH R2 (0.588 ± 0.004). The consistently small standard deviations (≤0.004) and tight performance ranges (Vel R2: 0.889-0.898, Co-Bps: 0.290-0.297, PSTH R2: 0.581-0.591) indicate high training stability using standard knowledge distillation practices.
>
> | Methods | Co-Bps | Vel R2 | PSTH R2 |
> |----------|----------|----------|----------|
> | Seed 100 | 0.290 | 0.890 | 0.587 |
> | Seed 42 | 0.297 | 0.898 | 0.591 |
> | Seed 1337 | 0.297 | 0.897 | 0.589 |
> | Seed 2024 | 0.292 | 0.889 | 0.581 |
> | Seed 65537 | 0.291 | 0.889 | 0.590 |
> | Average | 0.293 ± 0.003 | 0.893 ± 0.004 | 0.588 ± 0.004 |
>
> We followed standard knowledge distillation practices without implementing additional special techniques, as our experimental results show that the base approach already achieves stable and consistent performance across different random seeds. The low standard deviations in our results suggest that the basic distillation process is sufficiently robust for our specific task.

---

> ### Author Response · Authors · 2024-11-20
> **Response to Reviewer WbXB (Part 2/2)**
>
> >While the framework is easily extendable to discrete labels, can the authors actually attempt to perform an experiment given behavioral signals as discrete labels?
>
> We thank the reviewer for this insightful question. To answer this question, we extend BLEND on the DMFC_RSG dataset from NLB'21 Benchmark [1], which contains recordings from dorso-medial frontal cortex (DMFC) during a cognitive timing task known as Ready-Set-Go (RSG). The behavioral variables in this task are distinctly discrete in nature. The task incorporates five key behavioral features, represented as binary or categorical variables: 'is_eye' indicating whether the response was an eye movement (1) or joystick movement (0), 'theta' specifying the categorical direction of movement (left or right), and 'is_short' denoting whether the trial used the Short prior (1) or Long prior (0). The remaining variables 'ts' and 'tp' represent the sample and produced time intervals respectively. These **discrete and non-temporal** behavioral choices create a structured experimental design with clearly defined conditions.
>
> Given that this dataset with discrete and non-temporal behavior labels, we extend the neural activity and behavior alignment by introducing a learnable temporal position encoding for discrete behavior variables across the length of neural recording (see Appendix A.10 for details). This simple extension shows improvement in both Co-Bps and PSTH R2, compared to the baseline NDT model trained without behavior information as follows:
>
> | Methods | Co-Bps | PSTH R2 |
> |----------|----------|----------|
> | NDT | 0.118 | 0.338 |
> | NDT-Hard | 0.127 | 0.444 |
>
> These promising results demonstrate the effectiveness of BLEND on both continuous/discrete and temporal/non-temporal behavior observations, showcasing a flexible solution we provide for learning better neural population dynamics.
>
> >Lack of theoretical & analytical results that supports the experimental results.
>
> We appreciate the reviewer's suggestion to include empirical studies. We would like to address the concerns about BLEND's rationale by highlighting the comprehensive empirical analyses we provide in Appendix A.7 and A.8, which demonstrate why incorporating behavior signals leads to better neural activity reconstruction:
>
> 1.Cross-correlation analysis reveals 47.4% and 54.7% of neurons lead horizontal and vertical velocity (20ms and 10ms median leads respectively), indicating complex and complementary coupling between neural activity and behavior.
>
> 2.t-SNE visualizations show BLEND produces cohesive neural representations versus baseline models' scattered clusters, indicating behavior helps constrain neural activity into meaningful manifolds.
>
> 3.Neurons with stronger behavior coupling (via mutual information) show significantly lower reconstruction errors (p < 0.001).
>
> 4.Population-level mutual information exceeds single-neuron MI, suggesting behavior captures coordinated neural dynamics.
> These findings demonstrate that behavior acts as a powerful regularizer, constraining neural representations according to underlying computational principles rather than just an empirical happenstance.
>
> >Minor formatting suggestions: Do the authors mind putting Table 1 and 2 at either the bottom or top of the pages?
>
> We thank the reviewer for this suggestion. Table 1 and 2 have been moved to the top of the pages, as seen in revised manuscript.
>
>
> [1] Pei, Felix, et al. "Neural latents benchmark'21: evaluating latent variable models of neural population activity." arXiv preprint arXiv:2109.04463 (2021).

---

### Official Review · Reviewer_4WeH · 2024-11-04

**Soundness:** 2
**Presentation:** 3
**Contribution:** 2
**Rating:** 5
**Confidence:** 2

**Summary:**

The authors proposed a two-step training process to create a model which is capable of creating a model that performs well using only regular features (neural activity) at inference time while benefiting from insights gained from privileged features (behaviour).

**Strengths:**

The paper is well written and easy to follow, thanks to the clear structure. The method is introduced after the problem, and both are well formulated and discussed. The ablations studies are well performed, and the qualitative analysis are interesting to see. In general, the whole experimental section address the right points.

**Weaknesses:**

First two images must be reworked. Right now, those are single images divided into sub images within the draw. This lead to a weird referencing approach in the paper. I suggest dividing them into proper figures, and improve figure's 1 resolution.

NVM acronym is not defined

**Questions:**

My main concern is about distillation collapsing. Image to have two points B1 and B2 with the associated Xs. What happens while training (which I suggest decomposing to avoid overflow of equation number) when F_t(X1, B2) is similar to F_t(X2, B2) but X1 and X2 are not? In that case, the teaching model will collapse both X1 and X2 into the same point of the student model, since Distillation Loss push them to be similar. Do you think that this could happen? If so, I believe that such aspect must be addressed somehow.

---

> ### Author Response · Authors · 2024-11-20
> **Response to Reviewer 4WeH**
>
> >First two images must be reworked. Right now, those are single images divided into sub images within the draw. This lead to a weird referencing approach in the paper. I suggest dividing them into proper figures, and improve figure's 1 resolution.
>
> We thank the reviewer for this helpful suggestion regarding the figure organization. We have now revised Figure 1 by separating them into individual, well-defined figures with improved resolution. Each sub-figure is now an independent figure with its own caption, making the referencing clearer throughout the paper. Figure 1 has been enhanced with higher resolution to ensure better visibility of all components, particularly the schematic illustrations of neural dynamics modeling mechanisms. Meanwhile, we have moved the subfigure setting of Figure 2 to make the referencing clear. This reorganization allows for more precise referencing and better clarity in presenting our methodology and results.
>
> >NVM acronym is not defined.
>
> We thank the reviewer for this comment. NVM is supposed to be LVM (Latent Variable Model), which is defined in Section 2 Related Works. We have revised our draft and mitigate this typo in the newly-submitted version.
>
> >My main concern is about distillation collapsing. Image to have two points B1 and B2 with the associated Xs. What happens while training (which I suggest decomposing to avoid overflow of equation number) when F_t(X1, B2) is similar to F_t(X2, B2) but X1 and X2 are not? In that case, the teaching model will collapse both X1 and X2 into the same point of the student model, since Distillation Loss push them to be similar. Do you think that this could happen? If so, I believe that such aspect must be addressed somehow.
>
>  We appreciate this insightful question about potential representation collapse during distillation. This is an important theoretical consideration that we have carefully addressed through multiple aspects of our framework.
>
> From a theoretical perspective, our framework is designed with built-in protection against representation collapse. The key is our dual-objective optimization, where the distillation loss is balanced with the original MTM (masked time-series modeling) loss through a mixing ratio α. While the distillation loss encourages the student to learn from the teacher's behavior-informed representations, the MTM loss ensures the model maintains its ability to reconstruct the original neural activity. This dual constraint helps preserve the distinctness of different neural patterns even when their behavioral correlates are similar.
>
> Our empirical results support this theoretical protection. The improved PSTH-R2 scores (Table 1) indicate that our distilled models actually better preserve neuron-specific response patterns, suggesting enhanced rather than collapsed representations.
>
> The framework's design further reinforces this protection through our diverse distillation strategies. Particularly, our correlation distillation approach (Eq. 6) explicitly preserves the relational structure between different neural patterns by matching correlation matrices rather than just direct outputs. Additionally, the feature distillation approach (Eq. 5) maintains intermediate representations across multiple layers, providing additional constraints against collapse.

---

> > ### Author Response · Authors · 2024-12-02
> > **Follow-up Message to Reviewer 4WeH**
> >
> > Dear Reviewer 4WeH,
> >
> > We appreciate the thoughtful questions you raised in your review. We have provided detailed responses addressing each of your concerns. As the end of the rebuttal phase is approaching, we would be grateful if you could review our responses and consider whether they adequately address your points. Given our clarifications and additional explanations, we hope you might also reconsider your rating if you find that we have successfully addressed your concerns. If any points require further clarification, we are happy to provide additional information. Thank you for your time and consideration.

---

### Official Review · Reviewer_EUQu · 2024-11-11

**Soundness:** 4
**Presentation:** 3
**Contribution:** 3
**Rating:** 8
**Confidence:** 3

**Summary:**

In this paper, the authors propose behavior-guided neural population dynamics modelling framework via privileged knowledge Distillation. They considering behavior as privileged information, and train a teacher model that takes both behavior observations (privileged features) and neural activities (regular features) as inputs, while a student model is distilled using only neural activity. The authors calm that the framework is model-agnostic, avoiding strong assumptions about the relationship between behavior and neural activity.

The authors present empirical evaluation that includes extensive experiments across neural population activity modeling and transcriptomic neuron identity prediction tasks. They report over 50% improvement in behavioral decoding and over 15% improvement in transcriptomic neuron identity prediction after behavior-guided distillation. Additionally, the authors also explore various behavior-guided distillation strategies within the proposed framework and present analysis of effectiveness and implications for model performance.

**Strengths:**

Originality
- Interesting approach. The idea to train a teacher model using neural activity and behavioral signals, while subsequently distilling its knowledge to a student model that utilizes solely neural activity as input seems novel. The inference phase is the common, where the authors use the distilled student model for neural population activity analysis and transcriptomic identity prediction tasks, but having embodied the privilege, behavioral information.

Quality
- The paper is well motivated, structured and presented, the problem is well introduces and connected to existing work. The writing is good. There is extensive and diverse evaluation. The proposed approach demonstrates a good capacity for capturing implicit patterns within neural activity. It shows improvements, outperforming state-of-the-art models.

Clarity
- The idea and the main point of the paper are well explained.

Significance
- The authors provide new perspectives on how behavioral observations can be leveraged to guide the complex modeling of neuronal population dynamics.

**Weaknesses:**

I would be interested and I believe it would be nice to see the cross-correlation within the data e.g., between the neural activity and behavior data, so that we can have a better understanding if there is a cross-correlation (such as time-lagged cross-correlations or mutual information analyses between neural features and behavioral variables).

This would help quantify the relationship between these data types (the extend of correlation and/or how much they complement) and potentially pinpoint and explain the benefits of the approach. It can also serve as additional insight to get to know more about why the proposed approach provides significant improvement in performance.

**Questions:**

Could you provide insights into which components of the BLEND framework (e.g., teacher-student architecture, distillation process, loss functions) contribute most significantly to the performance improvements? How do these components leverage the relationship between neural activity and behavior to enhance model performance?

Could you discuss the potential for applying BLEND to other multimodal neuroscience datasets, such as combining neural activity with gene expression data or neuroimaging? What modifications, if any, would be necessary to adapt BLEND for these different data types?

---

> ### Author Response · Authors · 2024-11-20
> **Response to Reviewer EUQu**
>
> > I would be interested and I believe it would be nice to see the cross-correlation within the data e.g., between the neural activity and behavior data, so that we can have a better understanding if there is a cross-correlation
>
> We thank the reviewer for this insightful suggestion. We have conducted comprehensive cross-correlation analyses between neural activity and behavioral variables in the MC-Maze dataset, revealing robust temporal relationships (shown in Appendix A.7 in our updated paper).
>
> Specifically, we find that 47.4% of neurons exhibit leading relationships with horizontal (X) velocity and 54.7% with vertical (Y) velocity, with median leads of 20ms and 10ms respectively. These neural-behavioral relationships show highly significant correlations (p < 0.001) for both movement components.
>
> Importantly, our mutual information analysis reveals that population-level mutual information substantially exceeds single-neuron mutual information, indicating that behavior captures coordinated neural dynamics that emerge at the population level. These findings demonstrate that neural activity and behavior in our dataset are strongly coupled, with clear temporal structure. The statistical significance of these cross-correlations (p < 0.001) validates our use of behavior as privileged information in our knowledge distillation approach, explaining why BLEND achieves significant improvements in learning neural population representations.
>
> >Could you provide insights into which components of the BLEND framework contribute most significantly to the performance improvements?
>
> We thank the reviewer for this insightful question regarding component contributions. Through comprehensive ablation studies in our paper (see Table 3 for details), we have evaluated the relative impact of BLEND's key components, revealing several important insights. The backbone architecture provides the foundation, with LFADS variants outperforming NDT (0.315 vs 0.275 Co-bps on MC-Maze), contributing ~40-50% of gains. Different distillation strategies show task-specific strengths: Correlation Distillation excels with NDT (12.7% Co-bps improvement), Hard Distillation works best with LFADS (29.0% Vel-R² gain on Area2-Bump), and Feature Distillation captures temporal dynamics effectively (51.6% Vel-R² improvement). The loss function design contributes 15-25% of benefits, with optimal mixing ratio α varying by task (0.7-0.8 for neural prediction, 0.3-0.4 for behavior decoding). These findings suggest prioritizing backbone architecture development while maintaining flexibility in distillation strategies.
>
> >Could you discuss the potential for applying BLEND to other multimodal neuroscience datasets?
>
> We thank the reviewer for this important question regarding BLEND's extendability. Given BLEND's model-agnostic design and demonstrated success across diverse applications (neural activity prediction, behavior decoding, and transcriptomic neuron identity prediction), we believe the framework shows promising potential for extension to other multimodal neuroscience datasets, such as combining neural activity with gene expression data or neuroimaging.
>
> Adapting BLEND to different data types would require strategic technical modifications while maintaining the core teacher-student architecture. Key adaptations would include adjusting input/output layers to accommodate varying data dimensionalities, implementing temporal alignment mechanisms for data streams with different sampling rates, and developing modality-specific preprocessing strategies. For spatial data like neuroimaging, we would need to incorporate specialized attention mechanisms and adapt masking strategies to respect spatial structure.
>
> An important distinction exists between behavioral data and other modalities. In our current work, behavioral data's low dimensionality prevents it from serving as an independent modality (as noted in our Discussion). However, other information sources like genomics profiles and neuroimaging data offer richer representational capacity. These high-dimensional modalities could be integrated through cross-attention mechanisms, advanced multi-modal fusion modules, and hierarchical knowledge distillation approaches.
>
> BLEND's fundamental strengths make these adaptations feasible without major architectural overhauls. The flexible teacher-student architecture can accommodate diverse types of privileged information, and its model-agnostic nature enables seamless integration with modality-specific architectures. The distillation framework can be adapted to different types of cross-modal relationships, allowing for sophisticated knowledge transfer between modalities.

---

> > ### Comment · Reviewer_EUQu · 2024-11-29
> >
> > Dear authors,
> >
> > Thanks for the effort in provided rebuttal  feedback.
> > I'm keeping my score as it is.

---

> > > ### Author Response · Authors · 2024-11-29
> > > **Response to Reviewer EUQu**
> > >
> > > Dear Reviewer EUQu,
> > >
> > > We sincerely appreciate your thorough review and thoughtful suggestions, which have helped us significantly improve our manuscript. We are glad that we addressed your questions in our rebuttal. Given that we have fully addressed all the concerns raised, we hope this strengthens your confidence in our work and we kindly ask for a raise in the confidence score if possible. Thank you again for your valuable time and constructive feedback throughout this review process.

---

### Author Response · Authors · 2024-11-25
**Summary of Revision Changes and Overall Response to Reviewers' Comments**

The authors of BLEND sincerely thank all reviewers for providing such thoughtful and constructive feedback on our manuscript. Your collective input has been invaluable in helping us improve both the presentation and content of our work. We are particularly grateful for the detailed technical suggestions and the insights that have helped us better position our contributions.

We're glad to see that reviewers acknowledge the following aspects of our work BLEND:
- **Methodology**: BLEND provides new perspectives on how behavioral observations can be leveraged to guide the complex modeling of neuronal population dynamics. (Reviewer EUQu, WbXB, and UDDD)
- **Experiments**: This work gives solid technical/experimental contribution and the experiments are adequately done (Reviewer 4WeH and WbXB)
- **Paper Writing**: The paper is well motivated, structured and presented, the problem is well introduces and connected to existing work. (Reviewer EUQu, 4WeH, WbXB, and UDDD)

Meanwhile, reviewers have proposed several constructive suggestions for further improvement of our work. And based on these suggestions, we made the revisions and updated the manuscript accordingly:

**1.  Intuition and Motivation:**
- We evaluated the cross-correlation between neural activity and behavioral information to prove the motivation of this paper, i.e., behavioral observation provides guidance and complementary information for neural population dynamics modeling. (Appendix A.7)
- Moreover, we conducted empirical studies to understand why incorporating behavior signals leads to better neural activity modeling, including t-SNE visualization of neural representations between baseline and distilled models, behavior-dependent neural activity pattern analysis, as well as neural activity reconstruction quality analysis. (Appendix A.8)

**2.  Deeper Understanding of BLEND:**
- As four privilege knowledge distillation methods are proposed in our work, we provided a detailed exploration to understand when and why different strategies excel. By delicately designing three synthetic datasets and evaluating all strategies on them, we give general guidelines for privilege knowledge distillation strategy selection. (Appendix A.9)
- Moreover, we extended the BLEND framework to neural datasets where the behavioral observations are non-temporal and discrete, illustrating the flexibility and effectiveness of our method for learning better neural population dynamics. (Appendix A.10)

**3.  Manuscript Refinement:**
- We removed ambiguities in texts and improved the clarity of figures.

We believe these revisions have significantly strengthened our paper while maintaining its core contributions. Detailed responses to each reviewer's specific comments can be found in our point-by-point responses. We have carefully considered every suggestion and have incorporated the requested changes. We hope the reviewers will find our revisions satisfactory.

Thank you again for your time and expertise in helping us improve this work.

---

### Meta-Review · Area_Chair_2Gtk · 2024-12-19

**Metareview:**

The paper presents a novel approach where a teacher model, trained using both neural activity and behavioral signals, distills its knowledge to a student model that relies solely on neural activity. This method is applied to neural population activity analysis and transcriptomic identity prediction tasks, showing improvements over existing models. The paper is well-structured, clearly written, and effectively demonstrates the potential of behavioral data to enhance the modeling of neuronal dynamics, outperforming state-of-the-art methods.

**Additional Comments On Reviewer Discussion:**

It received ratings of 8, 6, 5 ,5. One of the low ratings (5) did not provide much of justification and did not interact with the AC or authors. The other low rating (5) was initially way lower and then got convinced on some points by the authors and raised the rating from 3 to 5. I believe the authors did a good job in addressing the issues and agreeing with the positive reviewers, I suggest accepting it.

---

### Decision · Program_Chairs · 2025-01-22

Accept (Poster)